# BiScale-GTR: Fragment-Aware Graph Transformers for Multi-Scale Molecular Representation Learning

## Abstract

Fragment-level representations provide a natural way to capture recurring molecular substructures and reuse their learned representations across molecules. However, a shared fragment identity alone may not fully describe how a fragment is instantiated in a particular molecule, since the same fragment can exhibit different chemical behavior depending on its surrounding atomic environment. Effective fragment-based molecular learning therefore requires representations that are both reusable across molecules and sensitive to local atomic context. We introduce BiScale-GTR, a self-supervised molecular representation framework built around context-grounded shared fragment tokens. BiScale-GTR constructs a reusable graph Byte Pair Encoding (graph-BPE) vocabulary using Weisfeiler–Lehman (WL)-based fragment identity, chemical validity filtering, and recursive out-of-vocabulary (OOV) decomposition. Each shared fragment token is then grounded with atom-level GNN representations through atom-to-fragment pooling and gated fusion, allowing the same fragment identity to acquire context-dependent representations in different molecular environments. A structure-aware fragment Transformer performs global reasoning over these atom-grounded tokens, capturing reusable substructure identity, local chemical context, and long-range molecular dependencies. Experiments on MoleculeNet, PharmaBench, and the Long Range Graph Benchmark demonstrate strong performance across classification and regression tasks. Attribution analysis further shows that BiScale-GTR highlights chemically meaningful recurring motifs, providing interpretable links between molecular structure and predicted properties. *Code will be released upon acceptance.*

## 1 Introduction

Predicting molecular properties benefits from representations that can generalize across molecules while remaining sensitive to molecular context. Atom-level graph models preserve fine-grained chemical information by representing molecules as atoms and bonds, and have been widely used for molecular property prediction (Gilmer et al., 2017; Yang et al., 2019). However, when molecules are modeled purely at the atom level, recurring functional groups and substructures must be learned implicitly from repeated atom–bond patterns. This makes it difficult to directly reuse motif-level knowledge across molecules.

Fragment-level representations provide a natural intermediate abstraction between atoms and whole molecules. Overlapping decomposition methods (Wang et al., 2025; Mu et al., 2025) can expose useful local subgraphs within individual molecules, but the resulting fragments are not necessarily aligned across molecules. In contrast, predefined substructure descriptors such as molecular fingerprints (Rogers & Hahn, 2010; Durant et al., 2002; Bolton et al., 2008), rule-based fragmentation schemes (Lewell et al., 1998; Degen et al., 2008), and data-driven vocabulary-learning methods (Luong & Singh, 2023) can assign consistent identities to recurring molecular patterns. Such reusable identities allow repeated motifs to share parameters, accumulate statistical evidence during pretraining, and support attribution over recurring chemical substructures. Nevertheless, reusable fragment identity should be viewed as an abstraction rather than a complete molecular representation. Tokenization compresses atom-level features, attachment sites, inter-fragment bond environments, and local chemical constraints into a discrete vocabulary item. This compression can make learning less data-efficient, because the model must infer missing atom-level chemical details indirectly

from data rather than observing them explicitly. This motivates a central question: how can a model reuse shared fragment identities across molecules while preserving the atom-level chemical information needed to represent each occurrence?

Answering this question requires two coupled design choices. The first is how to construct a robust reusable fragment vocabulary. Molecular tokenization is not merely a preprocessing step, but a modeling decision that determines which chemical units can be represented, reused, and generalized to unseen molecules. Recent studies of molecular foundation-model tokenization show that closed-vocabulary SMILES tokenizers can obscure chemical information through unknown tokens, especially when rare elements, charges, chirality, or other bracketed-atom features are not covered (Wadell et al., 2026). Graph-based fragment methods such as GraphFP further show that data-driven subgraph vocabularies and graph tokenization can improve molecular representation learning (Luong & Singh, 2023). However, a reusable molecular token space requires more than frequent subgraph extraction: fragment tokens should be chemically valid, consistently identifiable across isomorphic occurrences, and robust to unseen subgraphs at inference time. These requirements motivate chemical validity filtering, permutation-invariant graph identity, and recursive decomposition of out-of-vocabulary fragments.

The second design choice is how to construct context-grounded fragment representations from reusable tokens. Existing atom–fragment models inject atom-level information into fragment representations through hierarchical atom–motif–graph message passing (Zang et al., 2023), atom- and motif-level GNNs with local atom-to-motif augmentation (Han et al., 2023), or multi-level message passing over atoms, bonds, fragments, and fragment connections (Panapitiya et al., 2026). These models demonstrate that atom-level evidence is useful for fragment-aware molecular representation. However, their atom–fragment coupling is molecule-specific rather than organized around reusable fragment token identities shared across molecules. Moreover, these models largely rely on GNN-based message passing, which captures long-range fragment interactions less directly than Transformer-style attention. In parallel, Graph Transformers have shown strong empirical performance by combining graph-structured inductive bias with attention-based interaction modeling (Wu et al., 2021; Rong et al., 2020; Mu et al., 2025). Among hybrid Graph Transformer architectures, parallel GNN–Transformer designs (Mu et al., 2025; Rampášek et al., 2022) are particularly relevant because they preserve separate local message-passing and global attention pathways before fusion, allowing local structural evidence and long-range interactions to be modeled complementarily.

We propose BiScale-GTR, a fragment-aware molecular representation framework that couples robust graph-based tokenization with atom-level contextual grounding. Inspired by the local–global separation in parallel GNN–Transformer designs, BiScale-GTR uses separate atom-level grounding and fragment-level reasoning pathways, but organizes both around reusable molecular fragment tokens. The tokenizer constructs a reusable fragment vocabulary through graph Byte Pair Encoding (graph-BPE), Weisfeiler–Lehman (WL)-based fragment identity (Weisfeiler & Leman, 1968), chemical validity filtering, and recursive decomposition of unseen fragments. This shared token table allows recurring substructures to reuse fragment-level knowledge across molecules. The architecture then grounds each shared fragment token in its local atomic environment. Specifically, a shallow GNN encodes atom-level chemical context through bond-constrained message passing, and atom representations are pooled into their corresponding fragment tokens and combined through gated fusion. A structure-aware fragment Transformer subsequently performs long-range reasoning over these atom-grounded fragment tokens. In this way, BiScale-GTR preserves reusable fragment identity while allowing each fragment occurrence to adapt to its molecule-specific atomic context. The overall framework is illustrated in Fig. 1.

Our contributions are summarized as follows:

- We propose BiScale-GTR, a fragment-aware Graph Transformer built around context-grounded shared fragment tokens, where reusable graph-BPE fragment tokens are grounded with atom-level GNN context before fragment-level Transformer reasoning.

- We introduce a robust graph-BPE molecular tokenizer with WL-based fragment identity, chemical validity filtering, and OOV decomposition, enabling a shared fragment-token vocabulary across molecules.

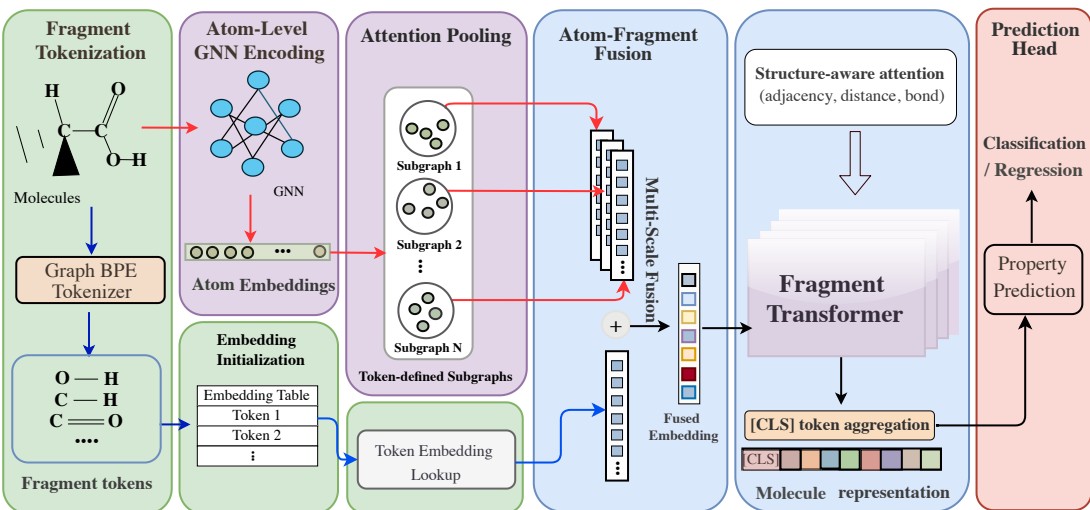

Figure 1: BiScale-GTR combines atom-level GNN encoding and fragment-level token representations. Multi-scale fusion integrates atom and fragment embeddings, which are modeled by a structure-aware fragment Transformer for molecular property prediction.

- We show that the shared fragment vocabulary supports chemically meaningful fragment-level interpretability by associating attribution scores with recurring molecular motifs.
- Experiments on MoleculeNet, PharmaBench, and LRGB demonstrate strong performance across classification and regression benchmarks.

## 2 Related works

BiScale-GTR connects three lines of work: molecular tokenization, atom–fragment representation learning, and Graph Transformer architectures. We review them separately to clarify the gap addressed by BiScale-GTR: reusable fragment tokens that are grounded by atom-level context and reasoned over with fragment-level attention.

### 2.1 Reusable Fragment Vocabularies and Molecular Tokenization

The emergence of molecular foundation models and Graph Transformer architectures has made the choice of molecular representation units a central design issue. Classical substructure representations, including descriptor-based fingerprints such as ECFP, MACCS keys, and PubChem fingerprints (Rogers & Hahn, 2010; Durant et al., 2002; Bolton et al., 2008), as well as rule-based fragmentation schemes such as RECAP and BRICS (Lewell et al., 1998; Degen et al., 2008), provide chemically meaningful substructure identities. However, their manually defined identities limit adaptation to data-specific recurring motifs. String-based substructure tokenizers such as SMILES Pair Encoding (SPE) (Li & Fourches, 2021) and Atom Pair Encoding (APE) (Leon et al., 2024) learn BPE-style merge rules over molecular strings, producing reusable string-level substructure tokens. However, Wadell et al. show that these atom-wise initialized tokenizers remain closed-vocabulary and can suffer from incomplete coverage, causing rare elements, isotopes, chirality, charges, and other bracketed-atom features to be obscured by unknown tokens (Wadell et al., 2026). They further note that string-level BPE tokenizers may conflate chemically distinct entities when textual merges are not chemically grounded (Wadell et al., 2026). These limitations motivate graph-fragment tokenizers with chemically grounded identities and fallback mechanisms for unseen structures.

Recent graph-based fragment methods move toward reusable fragment vocabularies learned directly from molecular graphs. GraphFP constructs fragments through principal subgraph mining for fragment-level

pretraining and finetuning (Luong & Singh, 2023), while GRAPHBPE (Shen & Póczos, 2024) and FragmentNet (Samanta et al., 2025) explore BPE-style graph tokenization by iteratively merging frequent graph substructures into reusable fragment tokens. These methods show that learned graph fragments can serve as intermediate units between atoms and whole molecules. Frequency-based fragment learning alone, however, does not fully address robust molecular tokenization. A reusable graph-fragment vocabulary should assign consistent identities to isomorphic occurrences, avoid chemically invalid or incoherent merges, and handle out-of-vocabulary fragments at inference time. To address these requirements, BiScale-GTR constructs a reusable fragment-token vocabulary with WL-based fragment identity for isomorphism-consistent matching, chemical validity filtering, and recursive OOV decomposition.

## 2.2 Atom–Fragment Grounding

Atom–fragment models study how fragment-level representations can be enriched with atom-level information inside individual molecular graphs. HiMol (Zang et al., 2023) augments each molecular graph with motif nodes and a graph-level node, then uses hierarchical GNN message passing across atom, motif, and graph levels. HimGNN (Han et al., 2023) constructs atom- and motif-level graphs and refines motif representations through local atom-to-motif augmentation. FragNet (Panapitiya et al., 2026) uses multi-level message passing over atoms, bonds, fragments, and fragment connections. These methods compute fragment representations through graph-specific message passing, showing that atom-level context is useful for fragment-aware molecular representation. However, because fragments are instantiated as nodes within each molecular graph, their identities are not organized as reusable vocabulary tokens shared across molecules. Thus, prior atom–fragment models ground molecule-specific fragments with atom-level information, whereas BiScale-GTR addresses a different setting: grounding shared fragment-token identities with occurrence-specific atomic context.

## 2.3 Graph Transformers for Fragment-Level Reasoning

Graph Transformers address the limitations of local message passing by extending self-attention to graph-structured data, enabling direct interactions between distant nodes. Since attention is structure-agnostic, graph topology is typically reintroduced through structural encodings. For example, Graphormer (Ying et al., 2021) injects node centrality, shortest-path distance, and edge features as attention biases. Hybrid GNN–Transformer architectures instead combine local message passing with global attention, and differ mainly in how the two are integrated (Rampášek et al., 2022; Chen et al., 2022). Sequential designs stack Transformer layers on GNN encoders, as in GraphTrans (Wu et al., 2021) and GROVER (Rong et al., 2020), so attention operates on representations already shaped by message passing. Interleaved designs alternate GNN and Transformer layers to refine local and global signals iteratively, as in TransGNN (Zhang et al., 2024). Parallel designs compute message passing and attention simultaneously before fusion, as in GraphGPS (Rampášek et al., 2022) and EHDGT (Mu et al., 2025), preserving complementary local and global pathways. This parallel modeling principle provides the architectural basis for BiScale-GTR. Since our goal is to realize context-grounded reusable fragment tokenization, we use local message passing to encode atom-level chemical context and global attention to model interactions among the resulting fragment tokens. In this design, the architecture is not proposed as a generic GNN–Transformer hybrid; rather, it serves to operationalize the two requirements identified above: robust shared fragment identities and occurrence-specific atom grounding.

# 3 Methods

In this section, we present BiScale-GTR, a self-supervised learning framework for molecular representation learning. We first introduce a BPE-based fragmentation strategy to construct chemically meaningful fragment tokens from molecular graphs. Next, we describe the BiScale-GTR architecture, which integrates an atom-level GNN with a fragment-level Transformer to jointly capture local chemical structures and long-range molecular dependencies. Finally, we present the self-supervised pretraining objectives and the fine-tuning procedure used for downstream molecular property prediction tasks.

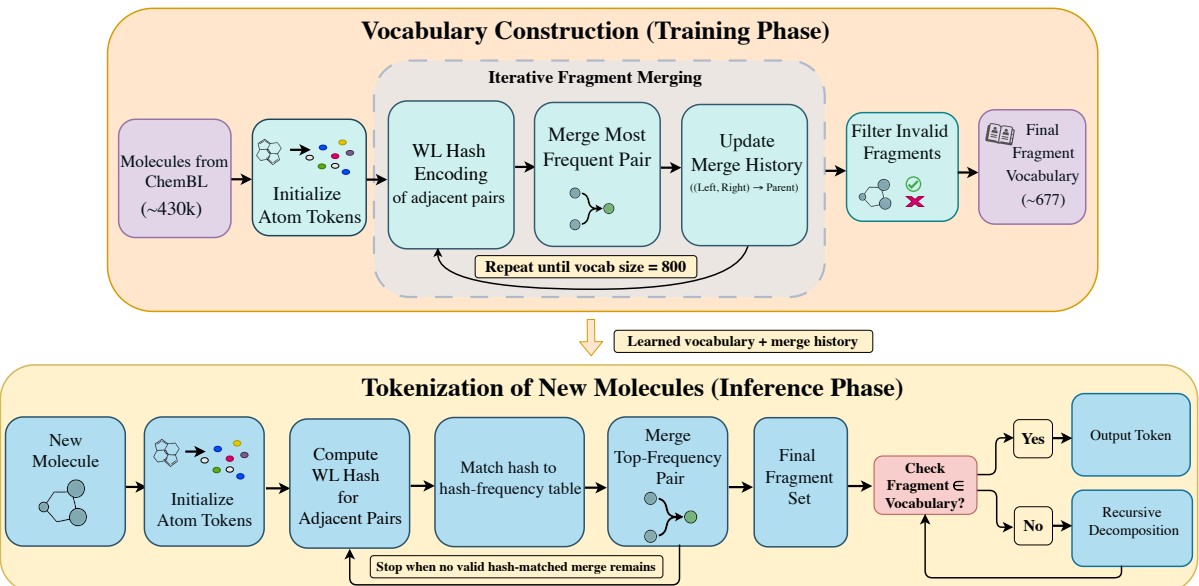

Figure 2: Overview of Graph BPE vocabulary construction and tokenization.

## 3.1 Fragment Token Construction

Motivated by the success of subword tokenization methods such as BPE in natural language processing (Shibata et al., 1999), we adapt an iterative merge-based procedure to molecular graphs to learn a fragment vocabulary. We construct the vocabulary from a processed subset of approximately 430K molecules from ChEMBL (Mayr et al., 2018), after removing duplicate molecules and invalid SMILES. Merging operates on induced subgraphs extracted from RDKit molecular structures. To ensure consistent fragment identification, we use WL graph hashing, where node labels encode atomic number and aromaticity and edge labels encode bond types. WL hashing maps isomorphic subgraphs to the same permutation-invariant identifier, enabling consistent fragment matching across molecules. The overall vocabulary construction and tokenization pipeline is illustrated in Fig. 2.

### 3.1.1 Fragment Vocabulary Construction

Each molecule is initialized at the atom level. At every iteration, candidate merges are enumerated between adjacent fragments to form larger induced subgraphs. For each candidate fragment, a WL hash is computed and used to aggregate fragment frequencies across the corpus. The fragment hash with the highest corpus frequency is then selected and merged across all molecules. The corresponding merge operations are recorded during vocabulary construction to form the merge history used for later recursive decomposition. The overall vocabulary construction procedure is summarized in Algorithm 1. Using this procedure, we construct an initial fragment vocabulary of 800 entries, following prior work showing that this size provides strong downstream performance while maintaining compact fragment graphs (Luong & Singh, 2023).

### 3.1.2 Fragment Validity Filtering

To prevent the fragment vocabulary from containing chemically implausible tokens, each candidate fragment is subjected to a set of chemical sanity checks. Specifically, we verify that the fragment forms a single connected component, satisfies relaxed valence constraints, and preserves aromatic ring integrity. In addition, we avoid fragments that break common functional groups by matching candidate fragments against a library of functional-group SMARTS patterns. Fragments that fail these checks are removed from the vocabulary. Starting from the initial vocabulary of 800 fragments, this filtering step eliminates chemically invalid motifs, resulting in a final vocabulary of 677 valid fragments. Additional statistics and analyses of the learned

---

**Algorithm 1:** Hash-Guided Graph BPE Vocabulary Construction

---

**Input:** Molecular corpus $\mathcal{C}$, target vocabulary size $V$
**Output:** Fragment vocabulary $\mathcal{V}$, merge history $\mathcal{T}$
Initialize $\mathcal{V}$ with atom tokens;
Initialize hash set $\mathcal{H}$ for fragments in $\mathcal{V}$;
Initialize merge history $\mathcal{T} \leftarrow \emptyset$;
**for** *each molecule $M \in \mathcal{C}$* **do**
   | Initialize the fragment partition of $M$ as individual atoms;

**while** $|\mathcal{V}| < V$ **do**
   | Initialize frequency table $F \leftarrow \emptyset$;
   | **for** *each molecule $M \in \mathcal{C}$* **do**
   |    | Enumerate all adjacent fragment pairs in $M$;
   |    | **for** *each adjacent pair producing merged fragment $f$* **do**
   |    |    | Compute WL hash $h(f)$;
   |    |    | Increment $F[h(f)]$;
   | Select the hash $h^*$ with the highest frequency in $F$;
   | **for** *each molecule $M \in \mathcal{C}$* **do**
   |    | Merge each adjacent fragment pair in $M$ whose merged fragment hashes to $h^*$;
   | Extract a representative merged fragment $f^*$ corresponding to $h^*$;
   | **if** $h(f^*) \notin \mathcal{H}$ **then**
   |    | Add fragment $f^*$ to vocabulary $\mathcal{V}$;
   |    | Add $h(f^*)$ to hash set $\mathcal{H}$;
   | Record merge rule $(f_a, f_b) \rightarrow f^*$ in $\mathcal{T}$;
Apply chemical validity filtering to $\mathcal{V}$;
**return** $\mathcal{V}, \mathcal{T}$;

---

fragment vocabulary, including fragment frequency coverage, fragment size distributions, vocabulary stability across vocabulary sizes, chemical diversity analysis, and vocabulary-size sensitivity studies, are provided in Appendix A.6. These results indicate diminishing returns beyond 800 fragments and support the use of the 800-fragment vocabulary in the main experiments.

### 3.1.3 Tokenization of New Molecules

Given the filtered fragment vocabulary and the recorded merge history, tokenization of a new molecule starts from atom-level tokens and iteratively considers adjacent fragment pairs. For each candidate merged fragment, a WL hash is computed and matched against the learned hash-frequency table. At each step, the candidate whose hash matches a learned fragment entry with the highest frequency is merged. This process is repeated until no adjacent pair produces a candidate hash that matches a learned fragment entry. After merging stops, fragments present in the filtered vocabulary are kept as tokens, while fragments not present in the filtered vocabulary are recursively decomposed using the stored BPE merge tree until a valid vocabulary fragment or an atom-level token is reached. If an atom type itself does not exist in the vocabulary, it is mapped to a special ⟨unk⟩ token. This mechanism guarantees deterministic tokenization and full coverage across diverse molecular inputs.

We quantify tokenizer coverage using the fallback rate,

$$r_{\text{fallback}} = \frac{N_{\text{fallback}}}{N_{\text{tokens}}} \tag{1}$$

which measures the fraction of tokens generated through recursive fallback decomposition during tokenization. Across the small-molecule benchmarks, fallback remains limited. For MoleculeNet datasets, fallback rates range from 0.29% to 12.48%, with most datasets lying between 3% and 9%. For PharmaBench datasets,

fallback rates are even lower, remaining around 0.6%–1.2%. In contrast, the long-range peptide datasets exhibit a higher fallback rate of approximately 26%, which is expected because the fragment vocabulary is learned from small-molecule structures in ChEMBL, whereas peptide datasets contain longer chains and substantially different structural patterns. This reflects a limitation of using a fixed ChEMBL-derived vocabulary, as structurally distinct molecular families may require more recursive fallback decomposition. Detailed tokenization statistics are provided in Appendix A.5. Vocabulary construction is performed once offline, and inference-time tokenization is lightweight relative to the model forward pass; detailed complexity analysis and empirical timing are provided in Appendix A.4.

## 3.2 Model Architecture

Let a molecular graph be denoted as $G = (V, E)$, where $V$ denotes the set of atoms and $E$ denotes the set of chemical bonds. Using the fragment vocabulary described in Section 3.1, each molecule is represented as a sequence of fragment tokens $T = t_i{}_{i=1}^m$. Our framework integrates atom-level structural representations with fragment-level contextual reasoning. The architecture consists of three components: (1) an atom-level graph encoder, (2) atom–fragment alignment and gated fusion, and (3) a fragment-level Transformer with structure-aware attention.

### 3.2.1 Atom-Level Graph Encoder

We first encode the molecular graph using an edge-aware Graph Isomorphism Network (GINE) (Hu et al., 2019). Each atom $\mathcal{V}_i$ is represented by a learnable embedding derived from its chemical attributes, including atomic number, chirality, and auxiliary chemical constraints (see Appendix A.11 for details). These features are projected into a $d$-dimensional embedding space. Bond features are incorporated through edge embeddings that encode bond type and bond direction. Following the GINE formulation, bond embeddings are added to neighbor messages during aggregation. Atom representations are iteratively updated through stacked message-passing layers:

$$h_i^{(l+1)} = \mathrm{MLP}^{(l)} \left( (1 + \epsilon) h_i^{(l)} + \sum_{j \in \mathcal{N}(i)} \mathrm{ReLU} \left( h_j^{(l)} + e_{ij} \right) \right), \tag{2}$$

where $\mathcal{N}(i)$ denotes the neighbors of atom $i$, $\epsilon$ is a learnable scalar parameter, and $e_{ij}$ denotes the learned edge embedding encoding bond type and bond direction for the bond connecting atoms $i$ and $j$. After $L$ layers, the encoder produces atom embeddings

$$H^{\mathrm{atom}} = h_i{}_{i=1}^{|V|},$$

which capture local chemical environments and bonding patterns. The architecture can operate under two complementary reasoning regimes. In the fragment-centric regime, the GNN operates on isolated fragment subgraphs extracted from the molecular graph, where each fragment is treated as an independent graph. In contrast, in the full-molecule regime, the GNN processes the entire molecular graph, preserving the original connectivity between fragments and enabling message passing across the full structure. In the experimental sections, we denote these two configurations as BiScale-GTR (Fragment) and BiScale-GTR (Molecule), respectively.

### 3.2.2 Fragment Representation via Atom Pooling

Each fragment $F_k$ corresponds to a set of atoms grouped by graph BPE tokenization. To derive fragment-level features from atom embeddings, we aggregate the atom representations belonging to the same fragment using attention pooling. Given the set of atoms $F_k$, the fragment representation is computed as

$$h_k^{\mathrm{frag}} = \sum_{i \in F_k} \alpha_i h_i,$$

where the attention weights are

$$\alpha_i = \frac{\exp(w^\top h_i)}{\sum_{j \in F_k} \exp(w^\top h_j)}.$$

This mechanism allows the model to emphasize chemically informative atoms within each fragment.

### 3.2.3 Atom–Fragment Alignment and Gated Fusion

Fragment tokens represent discrete subgraph patterns drawn from the learned fragment vocabulary, while pooled atom representations capture local structural information. To integrate these complementary signals, we fuse the pooled atom representations with the fragment token embeddings that initialize the Transformer input. Let $e_k = \text{Embedding}(t_k)$ denote the fragment token embedding and $\tilde{h}_k = W_a h_k^{\text{frag}}$ denote the aligned atom representation projected into the same embedding space. The fusion gate is computed as

$$g_k = \sigma\left(W_g[e_k; \tilde{h}_k]\right), \tag{3}$$

where $[;]$ denotes concatenation. The final fused fragment representation is

$$z_k = (1 - g_k)e_k + g_k\tilde{h}_k. \tag{4}$$

The fused representation $z_k$ is then used as the input to the fragment Transformer. This gating mechanism allows the model to adaptively balance fragment identity information and atom-level structural signals.

### 3.2.4 Fragment-Level Transformer

Given the fused fragment representations $Z = {z_k}_{k=1}^m$, where $m$ denotes the number of fragment tokens in the molecule, we apply a Transformer encoder to model long-range dependencies between fragments. The standard self-attention projections are computed as

$$Q = W_Q Z, \quad K = W_K Z, \quad V = W_V Z.$$

The attention logits incorporate structural biases derived from the fragment graph:

$$A = \text{softmax}\left(\frac{QK^\top}{\sqrt{d}} + B_{\text{graph}}\right).$$

Here $B_{\text{graph}}$ injects molecular topology into the attention mechanism and consists of three components:

$$B_{\text{graph}} = B_{\text{adj}} + B_{\text{dist}} + B_{\text{bond}}.$$

**Fragment connectivity bias.** The adjacency structure of the fragment graph provides a base connectivity signal. Connected fragment pairs receive a learnable bias, while non-connected pairs and diagonal self-attention entries receive a separate learnable bias initialized to zero. This encourages attention to adapt to fragment connectivity during training.

**Shortest-path distance bias.** To capture longer-range structural relationships, we incorporate shortest-path distance embeddings between fragment pairs. Let $d_{ij}$ denote the shortest-path distance between fragments $i$ and $j$ in the fragment graph. Distances are capped at 8, so all larger distances are mapped to the same bucket. The distance bias is obtained via an embedding lookup:

$$B_{\text{dist}}(i,j) = \text{Embedding}(\min(d_{ij}, 8)),$$

which is added to the attention logits for each attention head.

**Bond-type structural bias.** For adjacent fragment pairs, we encode the inter-fragment bond type and, when available, the RDKit-derived bond direction category. Let $b_{ij}^{\text{type}} \in {\text{single}, \text{double}, \text{triple}, \text{aromatic}}$ denote the bond type, and let $b_{ij}^{\text{dir}} \in {\text{none}, \text{end-up-right}, \text{end-down-right}}$ denote the bond direction category used in our implementation. For attention head $h$, the bond bias is computed as

$$B_{\text{bond}}(i,j,h) = E_{\text{type}}(b_{ij}^{\text{type}})_h + E_{\text{dir}}(b_{ij}^{\text{dir}})_h,$$

where $E_{\text{type}}$ and $E_{\text{dir}}$ are learnable embedding tables. For non-adjacent fragment pairs, we set $B_{\text{bond}}(i,j,h) = 0$. The fused fragment representations are processed by a Transformer encoder with the above structural biases. We prepend a learnable [CLS] token to the fragment sequence and use its final hidden state after $L$ Transformer layers as the molecule-level representation.

### 3.3 Training Objectives

During pretraining, we adopt a masked fragment prediction objective similar to masked language modeling (Devlin et al., 2019). A subset of fragment tokens is replaced with a mask token, and the model is trained to predict the original fragment identity based on the surrounding context. Masked positions are sampled using a frequency-aware strategy rather than uniform sampling. Let $f_i$ denote the global frequency of fragment $i$ in the pretraining dataset. The masking weight is defined as $w_i \propto 1/\sqrt{f_i}$. This increases the probability of masking less frequent fragments.

## 4 Experiments

### 4.1 Datasets

We evaluate our model on several widely used molecular property prediction benchmarks. For classification tasks, we use datasets from MoleculeNet (Wu et al., 2018). For regression tasks, we evaluate on the PharmaBench ADMET property prediction benchmark (Niu et al., 2024). In addition, we assess the model's ability to capture long-range structural dependencies using the Peptides-func and Peptides-struct datasets from the LRGB (Dwivedi et al., 2022).

**MoleculeNet**  For classification tasks, we evaluate our model on seven biological datasets from the MoleculeNet benchmark. MoleculeNet is a widely used benchmark suite for molecular machine learning that provides standardized datasets, evaluation metrics, and data splits, enabling consistent comparison between models. These datasets cover a diverse set of biochemical and toxicity-related prediction tasks and are commonly used to evaluate the ability of models to learn molecular representations for biological property prediction. We adopt scaffold splitting (Luong & Singh, 2023) to ensure that molecules in the training, validation, and test sets contain distinct molecular scaffolds.

**PharmaBench**  For regression tasks, we evaluate our model on the PharmaBench ADMET property prediction benchmark. PharmaBench is a curated benchmark dataset constructed from public bioassay data sources, containing experimentally measured ADMET properties relevant to drug discovery. Compared to earlier ADMET datasets, PharmaBench provides larger and more diverse datasets that better reflect compounds encountered in real-world drug discovery pipelines. We focus on nine regression datasets including CYP2C9, CYP2D6, CYP3A4, HLMC, MLMC, RLMC, LogD, PPB, and Sol. We follow the predefined scaffold split provided by PharmaBench, which partitions each dataset into training and test sets with a ratio of 4:1.

**LRGB**  We further evaluate our model on the peptide datasets from the LRGB, which are designed to assess a model's ability to capture long-range dependencies in molecular graphs. The benchmark includes two peptide datasets derived from 15,535 peptide molecular graphs: Peptides-func, a multi-label graph classification task with 10 functional classes, and Peptides-struct, a graph regression task predicting 5 structural properties derived from peptide 3D structures. Both datasets use the official split provided by LRGB, where the data is divided into 70% training, 15% validation, and 15% test sets. More information regarding the datasets is provided in Appendix A.1.

### 4.2 Experimental Configurations

During pretraining, 20% of fragment tokens are selected for masking using a frequency-based sampling strategy. The model is optimized using AdamW (Loshchilov & Hutter, 2017) with a learning rate of $4 \times 10^{-4}$ and a batch size of 256. Pretraining is performed for approximately 503k optimization steps. Atom-level structural representations are computed using a 3-layer GIN. The Transformer encoder consists of 6 layers with hidden dimension $d = 256$, 8 attention heads, and a feed-forward dimension of 1024. Dropout is set to 0.1. Fine-tuning is also performed using the AdamW optimizer. Batch size and weight decay are selected individually for each benchmark to accommodate differences in dataset size and task characteristics. Detailed hyperparameter settings for each dataset are provided in Appendix A.2. For classification tasks

with significant class imbalance, positive class weighting is applied during training. For the LRGB, models are trained for 200 epochs. For MoleculeNet and PharmaBench, we instead apply early stopping based on the target evaluation metric. The final model is evaluated on the test set using the checkpoint with the highest validation score. All experiments are conducted on a single NVIDIA RTX 4090 GPU. Each experiment is repeated with 10 different random seeds, and we report the mean and standard deviation of the evaluation metrics across runs. Empirical wall-clock and GPU-hour costs for vocabulary construction, pretraining, fine-tuning, and inference are reported in Appendix A.3.

## 4.3 Baselines

We compare BiScale-GTR with representative models from several major paradigms in molecular representation learning. For the MoleculeNet benchmark, we include self-supervised graph representation learning methods, including GraphMVP (Liu et al., 2021), GraphMAE (Hou et al., 2022), Mole-BERT (Xia et al., 2023), GraphFP (Luong & Singh, 2023), SimSGT (Liu et al., 2023), along with MORE (Son et al., 2025), a multi-level molecular pretraining framework, and GraphGPS+LAC (Yang et al., 2025), a GraphGPS-based architecture with auxiliary learning objectives. For the PharmaBench benchmark, we follow the baselines reported in the original PharmaBench paper (Niu et al., 2024), including classical machine learning models (Random Forest (Rigatti, 2017), XGBoost (Chen & Guestrin, 2016)), graph neural networks (CMPNN (Swanson, 2019), FP-GNN (Cai et al., 2022)), Transformer-based architectures (DHTNN (Song et al., 2023), Transformer-M (Luo et al., 2022)), large-scale pretraining approaches (MPG (Li et al., 2020), KANO (Li et al., 2023)), fragment-based models (FraGAT (Zhang et al., 2021), FragFormer (Wang et al., 2025)), heterogeneous graph models (PharmHGT (Jiang et al., 2023)), and the 3D molecular foundation model Uni-Mol (Zhou et al., 2023). For the long-range molecular datasets, we adopt the baselines reported in the LRGB benchmark, including local message passing GNNs (GCN (Kipf & Welling, 2016), GCNII (Chen et al., 2020), GINE (Hu et al., 2019), GatedGCN (Bresson & Laurent, 2017)) and graph Transformer models with structural encodings, such as LapPE-based Transformers (Dwivedi et al., 2023) and SAN (Kreuzer et al., 2021). We also include fragment-based approaches such as GraphFP and FragFormer for additional comparison.

## 5 Results and Discussions

We evaluate our model on both classification and regression tasks across multiple molecular benchmarks. Classification results are reported on seven MoleculeNet datasets and long-range graph classification tasks from the LRGB benchmark. Regression performance is evaluated on the PharmaBench benchmarks and long-range peptide datasets.

### 5.1 Evaluation on MoleculeNet

To study the impact of GNN message-passing scope, we evaluate two variants of BiScale-GTR that differ in the scope of the GNN encoder. In BiScale-GTR (Fragment), the GNN operates on isolated tokenizer-defined fragment subgraphs. In BiScale-GTR (Molecule), the GNN processes the entire molecular graph before interacting with the Transformer.

Table 1 reports ROC-AUC results on seven MoleculeNet classification benchmarks. Overall, BiScale-GTR (Molecule) achieves the best performance on four out of seven datasets, demonstrating strong performance across diverse molecular property prediction tasks. Of note, MUV and HIV are highly imbalanced datasets with negative-to-positive ratios of approximately 500:1 and 11:1, respectively, and BiScale-GTR (Molecule) achieves the best ROC-AUC on both datasets, indicating that the proposed framework remains robust under severe class imbalance. BiScale-GTR (Fragment) achieves the best performance on BBBP, suggesting that preserving localized fragment semantics may be beneficial when predictive signals are concentrated in local structural motifs. A similar trend is observed on ToxCast, where the fragment variant outperforms the molecule-level model. In contrast, on most other datasets, including Tox21, MUV, BACE, SIDER, and HIV, the molecule regime performs better, suggesting that context-aware fragment representations are often beneficial for downstream prediction.

Compared with recent self-supervised baselines, BiScale-GTR achieves competitive or superior results while maintaining strong data efficiency. In particular, SimSGT shows competitive performance on HIV, MUV, and Tox21 but relies on substantially larger pretraining corpora (2M molecules), whereas our model is pretrained on only 430K molecules. Compared to the GNN-based framework GraphFP, which is pretrained on the same dataset as ours, BiScale-GTR (Molecule) outperforms GraphFP on five out of seven datasets, while BiScale-GTR (Fragment) achieves better performance on BBBP, demonstrating the performance of the proposed Transformer–GNN architecture.

These observations suggest that different datasets may benefit from different levels of structural context, ranging from localized fragment-level information to broader molecular context. Based on its stronger overall performance, we use BiScale-GTR (Molecule) as the default full model in subsequent experiments, while BiScale-GTR (Fragment) serves as a diagnostic variant for analyzing the local-versus-contextual representation trade-off. To better understand this behavior, we perform a two-regime analysis in Section 5.6.

Table 1: ROC-AUC (%) comparison on MoleculeNet biological classification tasks. Results are reported as mean $\pm$ standard deviation when available. Baseline results are taken directly from the corresponding original papers using the same data split protocol. **Bold** indicates the best result and underlined values denote the second-best performance.

| Model | Pretrain Data Size | BBBP | Tox21 | MUV | BACE | ToxCast | SIDER | HIV |
|---|---|---|---|---|---|---|---|---|
| GraphMVP (Liu et al., 2021) | 50k | 70.8±0.5 | 74.9±0.8 | 77.7±0.6 | 79.3±1.5 | 63.1±0.2 | 60.2±1.1 | 76.0±0.1 |
| GraphMAE (Hou et al., 2022) | 2M | 71.2±1.0 | 75.2±0.9 | 76.4±2.0 | 78.2±1.5 | 63.6±0.3 | 60.5±1.2 | 76.8±0.6 |
| Mole-BERT (Xia et al., 2023) | 2M | 71.9±0.8 | **76.8±0.5** | 78.9±1.8 | 80.8±1.4 | 64.3±0.2 | 62.8±1.1 | 78.2±0.8 |
| SimSGT (Liu et al., 2023) | 2M | 72.2±0.9 | 76.8±0.9 | 81.4±1.4 | 84.3±0.6 | **65.9±0.8** | 61.7±0.8 | 78.0±1.9 |
| GraphFP (Luong & Singh, 2023) | 430k | 72.0±1.7 | 74.0±0.7 | 75.4±1.9 | 80.5±1.8 | 63.9±0.9 | 63.6±1.2 | 78.0±1.5 |
| MORE (Son et al., 2025) | 2M | 71.9±0.9 | 75.6±0.5 | – | 82.8±1.3 | 64.6±0.6 | 60.9±0.6 | 77.0±0.7 |
| GraphGPS+LAC (Yang et al., 2025) | – | 73.6 | 74.0 | 71.3 | 82.5 | 73.7 | 60.4 | 77.6 |
| BiScale-GTR (Fragment) | 430k | **73.8±0.4** | 73.1±0.9 | 74.7±0.4 | 83.8±1.2 | 63.9±0.2 | 60.1±1.1 | 77.9±1.1 |
| BiScale-GTR (Molecule) | 430k | 68.4±0.8 | 76.1±0.4 | **81.6±0.9** | **85.0±1.1** | 62.2±0.2 | **64.2±0.9** | **79.2±0.6** |

## 5.2 Evaluation on the PharmaBench

We show the results on PharmaBench regression tasks in Table 2. Overall, BiScale-GTR (Molecule) achieves the best performance on five out of nine tasks (CYP2C9, HLMC, MLMC, RLMC and PPB), demonstrating good generalization across diverse ADMET prediction tasks. The model shows particularly strong performance on microsomal clearance prediction tasks (HLMC, MLMC, and RLMC), where it achieves the lowest Root Mean Squared Error (RMSE) among all compared methods. These tasks require modeling complex interactions between multiple molecular substructures that influence metabolic stability. The improved performance suggests that our GNN–Transformer hybrid framework can capture both local chemical environments and broader structural dependencies relevant to metabolic processes. On the remaining datasets, compared to several fragment-aware methods, including GraphFP, FraGAT, PharmHGT, and FragFormer, BiScale-GTR consistently outperforms GraphFP and FraGAT across these tasks, and achieves performance close to the more advanced fragment-based architectures FragFormer and PharmHGT.

## 5.3 Evaluation on the LRGB Benchmark

We further evaluate BiScale-GTR on the LRGB. As shown in Table 3, BiScale-GTR achieves the best performance on peptides-func, reaching an Average Precision (AP) of 0.6717, outperforming all previous baselines including FragFormer. Although the improvement over FragFormer is modest, it is worth noting that FragFormer relies on a knowledge fusion layer that incorporates handcrafted molecular descriptors. As reported in the FragFormer paper, removing this knowledge fusion module reduces its performance to 0.6571 AP, highlighting the contribution of descriptor-based features. In contrast, BiScale-GTR achieves higher performance using only learned representations derived from the molecular graph and fragment structure. On Peptides-struct, BiScale-GTR achieves a Mean Absolute Error (MAE) of 0.2621, remaining competitive

Table 2: Performance comparison on PharmaBench regression tasks. Results are reported as RMSE (↓). Baseline results are taken directly from PharmaBench paper (Niu et al., 2024)

| Model | CYP2C9 | CYP2D6 | CYP3A4 | HLMC | MLMC | RLMC | LogD | PPB | Sol |
|---|---|---|---|---|---|---|---|---|---|
| RF (Rigatti, 2017) | 18.471 | 18.041 | 16.540 | 0.813 | 0.987 | 0.958 | 1.249 | 0.204 | 0.918 |
| XGBoost (Chen & Guestrin, 2016) | 17.582 | 17.819 | 16.123 | 0.647 | 0.844 | 0.819 | 1.071 | 0.186 | 0.832 |
| CMPNN (Swanson, 2019) | 18.377 | 19.156 | 16.701 | 0.921 | 1.130 | 0.939 | 0.807 | 0.236 | 0.858 |
| FPGNN (Cai et al., 2022) | 16.933 | 17.611 | 15.606 | 0.604 | 0.774 | 0.716 | 0.838 | 0.179 | 0.747 |
| DHTNN (Song et al., 2023) | 17.449 | 17.890 | 16.156 | 0.729 | 0.926 | 0.915 | 0.912 | 0.235 | 0.828 |
| KANO (Li et al., 2023) | 17.350 | 17.622 | 15.307 | 0.554 | 0.767 | 0.762 | 0.766 | 0.185 | 0.772 |
| MPG (Li et al., 2020) | 17.417 | 17.527 | **14.376** | 0.541 | 0.723 | 0.685 | 0.758 | 0.170 | 0.758 |
| UniMol (Zhou et al., 2023) | 17.774 | 18.071 | 15.895 | 0.613 | 0.824 | 0.651 | 0.745 | 0.179 | **0.707** |
| Trans-M (Luo et al., 2022) | 18.080 | 17.677 | 15.867 | 0.567 | 0.744 | 0.677 | 0.737 | 0.172 | 0.834 |
| KP-GPT (Wu et al., 2018) | 17.036 | 16.860 | 16.379 | 0.564 | 0.726 | 0.881 | 0.728 | 0.172 | 1.221 |
| GraphFP (Luong & Singh, 2023) | 17.367 | 21.183 | 17.219 | 0.764 | 0.878 | 0.771 | 0.835 | 0.208 | 1.935 |
| FraGAT (Zhang et al., 2021) | 17.788 | 22.503 | 20.313 | 0.775 | 0.849 | 1.050 | 0.945 | 0.220 | 1.352 |
| PharmHGT (Jiang et al., 2023) | 17.490 | 15.020 | 16.077 | 0.544 | 0.820 | 0.677 | 0.676 | 0.172 | 0.954 |
| FragFormer (Wang et al., 2025) | 16.855 | **14.425** | 15.894 | 0.514 | 0.702 | 0.596 | **0.667** | 0.157 | 0.895 |
| BiScale-GTR (Molecule) | **16.633** | 16.901 | 16.011 | **0.501** | **0.696** | **0.571** | 0.801 | **0.153** | 0.977 |

with strong Transformer-based baselines such as SAN and the LapPE-enhanced Transformer. Overall, these results suggest that GNN-based atom-level encoding and fragment-level Transformer reasoning provide complementary benefits: the former preserves fine-grained local chemical environments, while the latter models long-range interactions among higher-level fragments. Together, they form an effective mechanism for peptide property prediction.

Table 3: Performance comparison on peptide benchmarks. Average Precision (AP ↑) is reported for Peptides-func and Mean Absolute Error (MAE ↓) for Peptides-struct. Results are shown as mean ± standard deviation. Baseline results are taken directly from the LRGB benchmark

| Model | Peptides-func (AP ↑) | Peptides-struct (MAE ↓) |
|---|---|---|
| GCN (Kipf & Welling, 2016) | 0.5930±0.0023 | 0.3496±0.0013 |
| GCNII (Chen et al., 2020) | 0.5543±0.0078 | 0.3471±0.0010 |
| GINE (Hu et al., 2019) | 0.5498±0.0079 | 0.3547±0.0045 |
| GatedGCN (Bresson & Laurent, 2017) | 0.5864±0.0077 | 0.3420±0.0013 |
| GatedGCN+RWSE (Dwivedi et al., 2022) | 0.6069±0.0035 | 0.3357±0.0006 |
| Transformer (Vaswani et al., 2017) + LapPE (Dwivedi et al., 2023) | 0.6326±0.0126 | **0.2529±0.0016** |
| SAN (Kreuzer et al., 2021) + LapPE | 0.6384±0.0121 | 0.2683±0.0043 |
| SAN + RWSE | 0.6439±0.0075 | 0.2545±0.0012 |
| GraphFP (Luong & Singh, 2023) | 0.6267±0.0073 | 0.3137±0.0019 |
| FragFormer (Wang et al., 2025) | 0.6693±0.0154 | − |
| BiScale-GTR (Molecule) | **0.6717±0.0107** | 0.2621±0.0022 |

## 5.4 Ablation Studies

In this section, we evaluate the impact of key components in BiScale-GTR to understand their contributions to molecular representation learning. All ablation variants are re-pretrained and fine-tuned using the same configurations as the full model. All results are averaged over three runs with different random seeds, and we report the mean and standard deviation.

**Component-wise analysis of BiScale-GTR.** To investigate the contribution of each component in BiScale-GTR, we conduct architecture ablation studies on MoleculeNet benchmarks, as shown in Table 4. The full model consistently achieves the best performance across all datasets, demonstrating the effectiveness of combining GNN and Transformer representations. Removing the GNN (Transformer-only) leads to a noticeable performance drop on most datasets, particularly on MUV (81.6 vs. 71.6) and BACE (85.0

vs. 65.0), indicating that local structural information captured by the GNN is critical for molecular representation learning. Since the Transformer-only variant retains reusable fragment tokens and fragment-level self-attention but removes atom-level GNN grounding, this degradation suggests that fragment-level context alone does not fully capture occurrence-specific local chemical environments. Conversely, the GNN-only variant performs substantially worse across all tasks, suggesting that relying solely on local message passing is insufficient to capture long-range dependencies and global context. Furthermore, replacing the adaptive fusion gate with fixed-sum fusion leads to lower performance, indicating that adaptively balancing transferable fragment-token identity and atom-derived local context is more effective than treating the two sources of information as equally additive. Overall, these results demonstrate that atom-level grounding and fragment-level Transformer reasoning provide complementary information, and their interaction through the adaptive fusion gate is important for achieving optimal performance.

Table 4: Architecture and fusion ablation on MoleculeNet datasets (ROC-AUC %). The fixed-sum variant replaces the adaptive gate with $z_k = e_k + \tilde{h}_k$.

| Variant | BBBP | Tox21 | MUV | BACE | ToxCast | SIDER | HIV |
|---|---|---|---|---|---|---|---|
| Full model (GNN + Transformer) | 68.4±0.8 | 76.1±0.4 | 81.6±0.9 | 85.0±1.1 | 62.2±0.2 | 64.2±0.9 | 79.2±0.6 |
| Transformer-only (w/o GNN) | 62.8±0.8 | 73.2±0.5 | 71.6±0.8 | 65.0±0.9 | 61.1±0.7 | 59.8±0.5 | 77.1±0.4 |
| GNN-only (w/o Transformer) | 58.7±0.3 | 51.9±0.5 | 51.8±0.3 | 59.2±1.1 | 50.2±0.3 | 53.3±0.4 | 51.0±0.5 |
| Fixed sum fusion | 67.4±0.8 | 75.1±0.6 | 78.1±0.5 | 77.0±1.0 | 60.8±0.3 | 62.2±1.1 | 78.4±0.5 |

**Tokenizer Ablation Study.**

**(1) Tokenizer ablation protocol.** To isolate the effect of molecular tokenization, we compare four tokenizer variants while keeping the BiScale-GTR architecture, pretraining corpus, masked-fragment objective, optimization schedule, and downstream fine-tuning protocol identical to the main experiments. These tokenizers represent different levels of molecular abstraction: atom-level tokenization treats each atom as an individual token; BRICS provides rule-based chemical fragments; Graph-BPE (SMILES) serves as a data-driven Graph-BPE baseline that uses canonical fragment SMILES for fragment matching, but does not apply chemical validity filtering or recursive OOV decomposition; and Graph-BPE + WL uses the proposed graph-aware tokenizer as described in Section 3.1. The BRICS, Graph-BPE (SMILES), and Graph-BPE + WL vocabularies are constructed from the same ChEMBL pretraining corpus, while the atom vocabulary is defined by atom types, resulting in 22 atom tokens, 674 BRICS tokens, 800 Graph-BPE (SMILES) tokens, and 677 Graph-BPE + WL tokens. For each tokenizer, we regenerate both the pretraining and downstream inputs using that tokenizer, ensuring that each model is trained and evaluated on its corresponding tokenized representation. Results are averaged over three random seeds.

**(2) Tokenizer Comparison** Table 5 shows that fragment-level tokenization generally outperforms atom-level tokenization, especially on BBBP, BACE, SIDER, and Peptide-func. The gap is particularly large on Peptide-func, where atom tokenization represents long peptide chains as many individual atoms and requires the model to infer recurring backbone and side-chain motifs from fine-grained sequences. In contrast, fragment-based tokenizers expose larger substructure units directly, reducing the effective sequence length and making repeated local motifs and long-range fragment interactions easier to model. Among fragment tokenizers, Graph-BPE + WL achieves the strongest overall performance, with clear gains over BRICS on MUV, BACE, HIV, and Peptide-func, suggesting that learned graph-based fragments capture useful motifs beyond fixed rule-based disconnections. Graph-BPE + WL also consistently outperforms Graph-BPE (SMILES), indicating that data-driven merging alone is insufficient without graph-invariant fragment identity, chemical validity filtering, and OOV handling. Overall, these results support learned graph-aware fragment tokens as an effective inductive bias under the same architecture and training protocol.

**Effect of pretraining and masking ratio.** We study the impact of pretraining and masking ratio on downstream performance, as shown in Table 6. Removing pretraining leads to a substantial performance drop across all datasets, confirming that the proposed pretraining strategy provides strong initialization and improves generalization.

Table 5: Tokenizer ablation on MoleculeNet and Peptide-func benchmarks. All models use the same BiScale-GTR architecture, pretraining corpus, and optimization settings; only the tokenizer differs. Results are reported as ROC-AUC (%) for MoleculeNet datasets and AP (%) for Peptide-func, averaged over three random seeds.

| Tokenizer | BBBP | Tox21 | MUV | BACE | ToxCast | SIDER | HIV | Peptide-func |
|---|---|---|---|---|---|---|---|---|
| Atom | 52.6±10.2 | 74.5±0.3 | 76.7±1.8 | 71.5±13.6 | 61.9±0.2 | 55.1±1.1 | 75.4±0.2 | 42.2±0.03 |
| BRICS | 68.0±1.4 | 75.0±0.3 | 73.0±1.7 | 79.1±2.2 | 62.0±0.1 | 62.7±0.4 | 75.1±0.6 | 62.5±0.01 |
| Graph-BPE (SMILES) | 63.4±0.7 | 73.4±0.1 | 76.1±0.5 | 79.3±2.5 | 60.9±0.6 | 60.1±0.7 | 76.4±1.1 | 58.8±0.03 |
| Graph-BPE + WL (Ours) | **68.4±0.8** | **76.1±0.4** | **81.6±0.9** | **85.0±1.1** | **62.2±0.2** | **64.2±0.9** | **79.2±0.6** | **67.2±0.01** |

We further analyze the effect of different masking ratios. A moderate masking ratio of 0.2 consistently achieves the best or near-best performance across most datasets, indicating a good balance between learning informative context and maintaining sufficient input signal. A lower masking ratio (0.1) results in slightly weaker performance, suggesting limited difficulty in the pretraining task, while a higher masking ratio (0.3) leads to performance degradation, likely due to excessive information removal.

Table 6: Effect of pretraining and masking ratio (ROC-AUC %).

| Variant | BBBP | Tox21 | MUV | BACE | ToxCast | SIDER | HIV |
|---|---|---|---|---|---|---|---|
| w/o pretraining | 59.8±0.3 | 70.5±0.8 | 59.9±0.6 | 66.2±1.3 | 59.8±0.3 | 59.6±0.7 | 74.6±0.4 |
| mask ratio = 0.1 | 69.3±0.5 | 74.3±0.7 | 77.9±0.5 | 83.2±0.8 | 61.3±0.4 | 62.1±0.6 | 77.9±0.3 |
| mask ratio = 0.2 (Full model) | 68.4±0.8 | 76.1±0.4 | 81.6±0.9 | 85.0±1.1 | 62.2±0.2 | 64.2±0.9 | 79.2±0.6 |
| mask ratio = 0.3 | 66.9±0.4 | 75.2±0.6 | 76.8±0.6 | 81.0±1.4 | 61.9±0.6 | 62.3±0.4 | 75.7±0.4 |

Additional ablations on tokenization refinements, structure-aware attention biases, GNN depth, and pre-training/masking behavior are provided in Appendices A.7–A.10.

## 5.5 Model Analysis and Interpretability

To interpret the fragment-level reasoning behavior of BiScale-GTR, we analyze model predictions using attention-based attribution and embedding visualization techniques. Because the fragment tokens come from a shared vocabulary, their attribution scores reflect the importance of recurring chemical motifs rather than molecule-specific fragments.

### 5.5.1 Fidelity evaluation and visualization

We estimate fragment-level importance using the attention rollout method (Abnar & Zuidema, 2020), which aggregates attention weights across Transformer layers to approximate each token's contribution to the final prediction. Since the Transformer operates on fragment tokens, the resulting attribution scores naturally correspond to fragment-level structural units. Detailed descriptions of the rollout procedure are provided in Appendix A.12. To evaluate the faithfulness of these attribution scores, we perform a fidelity test based on fragment removal. Fragments are ranked by their attribution scores, and the top-ranked fragments are removed from the input molecule. We then measure the change in the model prediction, where a larger decrease indicates higher attribution faithfulness. For visualization, fragment importance scores are mapped back to their corresponding atoms within the fragment to highlight important substructures on the molecular graph.

**Faithfulness evaluation.** We evaluate attribution faithfulness on four representative MoleculeNet datasets: HIV, Tox21, BACE, and BBBP. For HIV, Tox21, and BACE, we use BiScale-GTR (Molecule), which achieves stronger predictive performance on these datasets. For BBBP, we instead use BiScale-GTR (Fragment), since it performs better on this task. As shown in Table 7, removing the top-3 most important fragments consistently leads to a noticeably larger drop in ROC-AUC ($\Delta_{\text{top}}$) than removing the bottom-3

fragments ($\Delta_{bottom}$). For example, on HIV and BACE, $\Delta_{top}$ exceeds 0.28, while $\Delta_{bottom}$ remains close to 0.10, resulting in large gaps of 0.187 and 0.186, respectively. A similar trend is observed on Tox21, although with smaller magnitude. BBBP shows the same overall pattern, with a particularly large gap between $\Delta_{top}$ and $\Delta_{bottom}$.

These results indicate that the fragments identified as important by the model are indeed critical for prediction, as their removal significantly degrades performance. In contrast, removing low-importance fragments has a much smaller effect, further validating the selectivity of the attribution method.

Table 7: Faithfulness evaluation via fragment masking. $\Delta_{top}$ measures the ROC-AUC drop (%) after removing top-3 most important fragments, while $\Delta_{bottom}$ measures drop (%) after removing bottom-3 fragments.

| Dataset | Metric | $\Delta_{top}$ | $\Delta_{bottom}$ | Gap ($\Delta_{top} - \Delta_{bottom}$) |
|---------|--------|------|---------|------|
| HIV | ROC-AUC ↓ | 28.9 | 10.2 | 18.7 |
| Tox21 | ROC-AUC ↓ | 11.1 | 4.1 | 7.0 |
| BACE | ROC-AUC ↓ | 29.8 | 11.2 | 18.6 |
| BBBP | ROC-AUC ↓ | 37.9 | 10 | 27.9 |

**Visualization.** Figure 3 presents fragment-level attribution visualizations on representative molecules from the HIV and Tox21 datasets. These datasets are chosen as representative benchmarks because they include compounds with known functional groups associated with biological activity , making them suitable for assessing whether the model captures chemically meaningful substructures. The visualization shows that the proposed attention rollout method successfully highlights chemically meaningful substructures. For example, in the HIV example, the top-attributed fragments contain hydrazide groups, which are consistent with hydrazide-containing scaffolds previously studied as potential HIV-1 integrase inhibitors and predicted to participate in active-site interactions(Sechi et al., 2008). In the Tox21 example, the model assigns high attribution to a fragment containing an azo linkage (–N=N–). This is consistent with Kazius et al., who identified azo-type and aromatic azo substructures as toxicophores associated with Ames mutagenicity (Kazius et al., 2005). These results demonstrate that the model not only achieves strong predictive performance but also captures relevant functional groups aligned with known chemical and biological mechanisms, supporting the interpretability and reliability of the learned fragment representations.

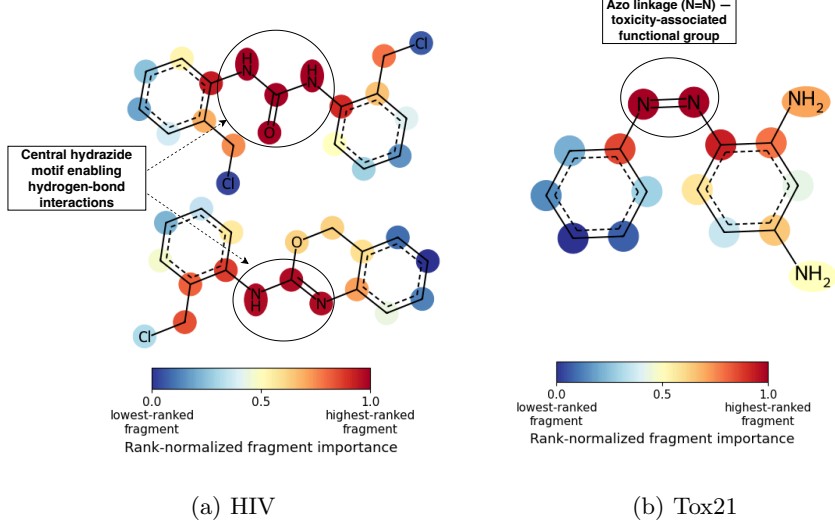

(a) HIV    (b) Tox21

Figure 3: Fragment-level attention attribution on representative molecules from the HIV and Tox21 datasets. Fragment colors show rank-normalized importance within each molecule based on CLS attention rollout scores. Circled substructures correspond to the highest-attribution learned fragment tokens selected by CLS attention rollout. Additional details are provided in Appendix A.13

### 5.5.2 Fragment embedding analysis

To further analyze the learned fragment representations, we visualize fragment embeddings using t-SNE. Figure 4 presents the two-dimensional t-SNE projection of the fragment embeddings. The displayed clusters arise from the t-SNE layout, while the color assigned to each fragment corresponds to its cluster membership derived from ECFP fingerprints, which capture underlying structural similarity between fragments. As shown in Figure 4(a), the Transformer without GNN produces fragment embeddings that are relatively dispersed, with weaker separation between clusters corresponding to different chemical substructures. Although some local grouping is observable, there is significant overlap between clusters, indicating limited alignment between the learned representations and chemical similarity. In contrast, Figure 4(b) suggests that incorporating the GNN leads to stronger within-cluster grouping of fragments with similar ECFP-derived structural labels. This improved organization suggests that the model more effectively captures chemically meaningful relationships between fragments. These observations are consistent with the higher normalized mutual information (NMI) score (see Appendix A.14) achieved by the Transformer with GNN, indicating stronger agreement between learned representations and fingerprint-based structural similarity. Overall, the results suggest that the GNN component enhances the Transformer's ability to encode structural information within its fragment-level embedding space.

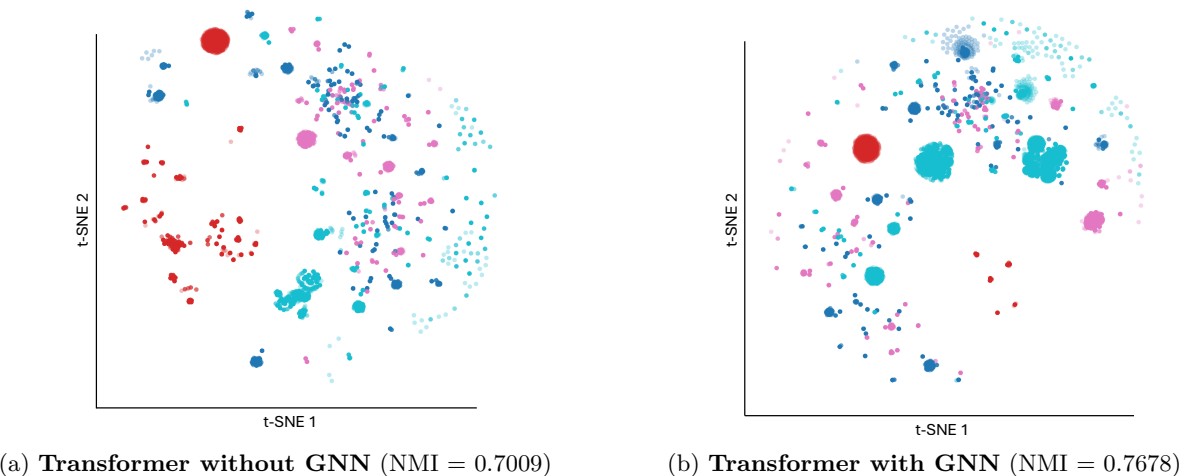

(a) **Transformer without GNN** (NMI = 0.7009)    (b) **Transformer with GNN** (NMI = 0.7678)

Figure 4: t-SNE visualization of fragment embeddings learned by the Transformer without (left) and with (right) the GNN encoder. Each point represents a fragment, and colors denote clusters derived from ECFP fingerprints. Fragments with similar embeddings are positioned closer together in the 2D t-SNE space. NMI measures the agreement between the embedding distribution and the ECFP-based clustering.

## 5.6 Two-Regime Analysis

To better understand why BiScale-GTR (Fragment) performs better on BBBP while BiScale-GTR (Molecule) is stronger on most other datasets, we analyze both the geometry of the learned token space and the concentration of fragment-level evidence. For the token-space analysis, we collect the final contextualized token representations on the BBBP test set, group them by token identity, and compute two statistics: (1) within-token spread, defined as the mean cosine distance from token occurrences to their centroid, and (2) centroid separation, defined as the mean pairwise cosine distance between token centroids. Lower within-token spread and higher centroid separation indicate a sharper and more discriminative token space.

As shown in Table 8, BiScale-GTR (Fragment) produces a substantially sharper token space than BiScale-GTR (Molecule), with both lower within-token spread and higher centroid separation. This suggests that encoding each tokenizer-defined fragment with a local fragment GNN preserves more discriminative subgraph semantics, whereas full-molecule message passing tends to smooth token representations by mixing information from the broader molecular context.

To compare fragment importance concentration across datasets with different baseline performance, we report the relative top-3 drop as

$$\text{Relative drop} = \frac{\Delta_{\text{top}}}{\text{original ROC-AUC}} \times 100\% \tag{5}$$

and visualize the results in Fig. 5. BBBP exhibits the largest relative ROC-AUC drop after removing the top-3 attributed fragments, indicating that its predictions are more strongly driven by a small number of highly important local motifs. Taken together, these results suggest that BBBP is a motif-driven task that benefits from a model capable of preserving sharper local token distinctions, which helps explain the advantage of BiScale-GTR (Fragment) on this dataset.

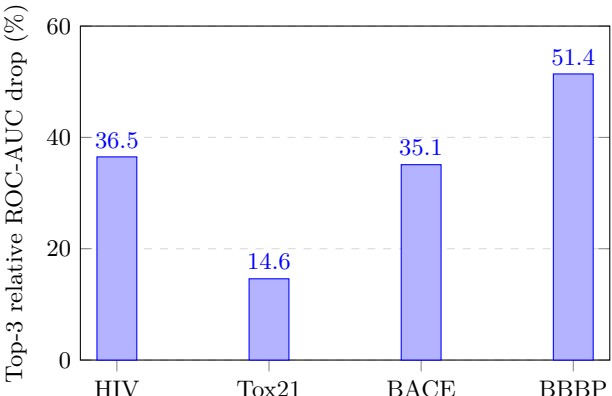

Figure 5: Top-3 relative ROC-AUC drop across datasets, computed using Eq. 5

Table 8: Token-space analysis on the BBBP test set. We report the average within-token spread and average centroid separation of the final contextualized token representations. Lower within-token spread and higher centroid separation indicate a sharper and more discriminative token space.

| Model | Within-token spread ($\downarrow$) | Centroid separation ($\uparrow$) |
|---|---|---|
| BiScale-GTR (Molecule) | 0.3627 | 0.4052 |
| BiScale-GTR (Fragment) | **0.2595** | **0.5549** |

## 6 Conclusions

In this work, we propose BiScale-GTR, a molecular representation framework built around context-grounded shared fragment tokens, where reusable graph-BPE fragment identities are grounded with atom-level message passing before fragment-level Transformer reasoning. A key component of our approach is Graph-BPE tokenization, a data-driven fragment vocabulary learning strategy that constructs chemically meaningful substructures through iterative graph merging while preserving structural validity and enabling robust fallback for unseen motifs. Built on this tokenization scheme, BiScale-GTR combines fragment-level representations with atom-level structural information through gated fusion and structure-aware attention biases, enabling the model to capture both fine-grained chemical environments and higher-level molecular patterns within a unified architecture. Extensive experiments across multiple molecular benchmarks demonstrate strong performance across diverse molecular property prediction tasks. These results highlight the importance of jointly modeling atom- and fragment-level representations, suggesting that multi-scale structural reasoning provides an effective inductive bias for molecular learning. BiScale-GTR focuses on the 2D molecular graph setting to study context-grounded reusable fragment tokens and isolate the effect of fragment-aware multi-scale representation learning. While recent studies (Feng et al., 2024; Gasteiger et al., 2020; Morehead & Cheng, 2024) show that 3D conformations can further improve molecular representation learning, especially for geometry-sensitive tasks, extending BiScale-GTR to 3D would require additional design choices such

as conformer generation, geometry-aware atom encoders, and geometry-aware atom-to-fragment grounding. We leave this direction to future work. More broadly, our Graph-BPE tokenizer may support large-scale molecular language modeling, and the proposed multi-scale architecture provides a foundation for incorporating additional structural levels as molecular datasets and conformation generation methods continue to advance.

## Acknowledgments

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

# A  Appendix

## A.1  Dataset Profiles

Dataset statistics and profiles used in our experiments are summarized in Tables 9, 10, and 11.

Table 9: Dataset profiles for PharmaBench benchmarks.

| Dataset | Size | Task Type | Description |
|---|---|---|---|
| CYP2C9 | 999 | Regression | Binding affinity to CYP2C9 |
| CYP2D6 | 1,214 | Regression | Binding affinity to CYP2D6 |
| CYP3A4 | 1,980 | Regression | Binding affinity to CYP3A4 |
| HLMC | 2,286 | Regression | Human liver microsomal clearance |
| MLMC | 1,403 | Regression | Mouse liver microsomal clearance |
| RLMC | 1,129 | Regression | Rat liver microsomal clearance |
| LogD | 13,068 | Regression | PH-adjusted lipophilicity |
| PPB | 1,262 | Regression | Plasma protein binding percentage |
| Sol | 11,701 | Regression | Water solubility |

Table 10: Dataset profiles for MoleculeNet classification benchmarks. Pos:Neg ratios indicate approximate class imbalance in each dataset.

| Dataset | Size | Tasks | Pos : Neg Ratio | Description |
|---|---|---|---|---|
| BBBP | 2,039 | 1 | $\sim 0.8 : 1$ | Blood-brain barrier permeability |
| Tox21 | 7,831 | 12 | varies $1$–$5 : 1$ | Toxicity on 12 biological targets |
| ToxCast | 8,575 | 617 | widely variable | High-throughput toxicity screening |
| SIDER | 1,427 | 27 | $1$–$15 : 1$ | Adverse drug reactions |
| MUV | 93,087 | 17 | $>500 : 1$ | Virtual screening validation |
| HIV | 41,127 | 1 | $\approx 11 : 1$ | HIV replication inhibition |
| BACE | 1,513 | 1 | $\sim 2 : 1$ | $\beta$-secretase 1 inhibition |

Table 11: The classification and regression tasks in the LRGB benchmark.

| | Peptides-func | Peptides-struct |
|---|---|---|
| Number of Graphs | 15,535 | 15,535 |
| Number of Tasks | 1 | 5 |
| Number of Classes | 10 | – |
| Task Type | Multi-label classification | Multi-label regression |
| Description | Peptides-func predicts peptide biological activities such as antibacterial and antiviral functions. Peptides-struct predicts global structural properties derived from peptide 3D conformations, including descriptors such as length and sphericity. | |

## A.2 Fine-tuning Configuration

**Two-stage fine-tuning strategy.** To stabilize the adaptation of pretrained molecular representations to downstream tasks, we adopt a two-stage fine-tuning strategy. In the first stage, the pretrained backbone is frozen and only the task-specific prediction head is optimized for a small number of epochs. This warm-up stage allows the classification or regression head to adapt to the downstream task without perturbing the pretrained representations. After the warm-up stage, selected components of the backbone are unfrozen and the model is jointly optimized. Specifically, we unfreeze the fragment attention pooling layer, the atom–fragment alignment module, the fusion gate, and the last few Transformer layers. Earlier layers of the backbone remain frozen to preserve general molecular representations learned during pretraining. Separate learning rates are used for the backbone and the task-specific head during this stage, with a smaller learning rate applied to the backbone parameters.

**Optimization settings.** All models are optimized using AdamW with different weight decay based on the benchmark characteristics. For MoleculeNet benchmarks, we use a batch size of 64 and a dropout rate of 0.1 or 0.2 (0.2 for MUV and HIV) depending on the susceptibility of each dataset to overfitting. For the PharmaBench benchmarks, we adopt a larger batch size and a learning-rate scheduler to stabilize training. For LRGB, a dropout rate of 0.05 is used based on validation performance. The complete optimization

configurations for each benchmark are provided in Table 12. For benchmarks with severe class imbalance (e.g., MUV), we compute task-specific positive class weights using the ratio between negative and positive samples in the training split and apply them in the binary cross-entropy loss.

Table 12: Fine-tuning hyperparameters for different benchmark groups. LR denotes learning rate.

| Benchmark | Batch Size | Weight Decay | Dropout | Head LR | Backbone LR | Scheduler (factor, patience) |
|---|---|---|---|---|---|---|
| MoleculeNet | 64 | $5 \times 10^{-5}$ | 0.10, 0.20 | $2 \times 10^{-4}$ | $5 \times 10^{-5}$ | None |
| PharmaBench | 256 | $1 \times 10^{-5}$ | 0.10 | $1 \times 10^{-3}$ | $1 \times 10^{-3}$ | Plateau (0.8, 6) |
| Long-range Peptides | 128 | $1 \times 10^{-5}$ | 0.05 | $3 \times 10^{-4}$ | $2 \times 10^{-4}$ | Plateau (0.5, 20) |

## A.3  Empirical Computational Cost

Table 13: Empirical computational cost of BiScale-GTR. GPU stages are measured on a single NVIDIA RTX 4090.

| Stage | Cost | Notes |
|---|---|---|
| Vocabulary construction, 800 fragments | 22 min wall-clock | One-time offline preprocessing |
| Pretraining | 8 GPU-hours | 503k optimization steps |
| Fine-tuning | 0.33–2.5 GPU-hours per dataset | Varies with dataset size and early stopping |
| Inference | ~15–25 ms/molecule | Tokenization + forward pass |

## A.4  Tokenization Complexity

Graph-BPE vocabulary construction is performed once offline. Each merge iteration scans unique adjacent fragment pairs across the corpus, computes WL hashes for candidate fragments, and selects the most frequent merge. For each candidate fragment, WL hashing costs (O(|f|r)), where (|f|) is the fragment size and (r) is the number of WL propagation rounds. Since molecular graphs have bounded degree, learned fragments are compact, and (r) is fixed, each merge iteration is linear in the total number of atoms. Thus, vocabulary construction has time complexity $O(KN\bar{n})$, where $K$ is the number of merge iterations, $N$ is the number of molecules in the construction corpus, and $\bar{n}$ is the average number of atoms per molecule. Chemical validity filtering is applied once offline to the learned vocabulary and is negligible relative to the main construction loop. The memory complexity is $O(N\bar{n} + K)$. For a new molecule with $n$ atoms, inference-time tokenization has worst-case time complexity $O(n^2)$ and memory complexity $O(n)$, but is practically lightweight for small molecules and negligible relative to the model forward pass. Empirical timing for this offline vocabulary construction step is provided in Appendix A.3.

## A.5  Tokenizer Coverage Across Datasets.

Table 14 reports tokenizer statistics across all downstream datasets, including the fallback rate and the UNK rate. The fallback rate measures the proportion of tokens generated through recursive decomposition when a fragment does not directly appear in the learned vocabulary, while the UNK rate indicates the fraction of tokens mapped to the [UNK] symbol. Across the MoleculeNet benchmarks, fallback rates remain relatively low, typically ranging between 0.3% and 12.5%, indicating that most molecular fragments can be directly represented by the learned fragment vocabulary. The UNK rate is consistently near zero, demonstrating that the tokenization scheme provides nearly complete coverage for small-molecule datasets. For the PharmaBench benchmarks, fallback rates are even lower, generally around 0.5%–1.5%, reflecting strong compatibility between the learned vocabulary and the chemical space represented in these datasets.

UNK rates remain negligible across all tasks, suggesting that the vocabulary effectively captures the majority of recurring molecular substructures. In contrast, the long-range peptide dataset exhibits a substantially higher fallback rate (26.97%). This is expected because the fragment vocabulary is constructed primarily from small-molecule structures in ChEMBL, whereas peptide datasets contain larger and structurally distinct motifs that are less frequently observed in the training corpus. Despite this distribution shift, the UNK rate remains zero, indicating that recursive decomposition successfully resolves unseen fragments into known substructures.

| Dataset | Fallback Rate | UNK Rate |
|---|---|---|
| *MoleculeNet* | | |
| BACE | 0.1248 | 0.0000 |
| BBBP | 0.0525 | 0.0022 |
| Tox21 | 0.0742 | 0.0033 |
| ToxCast | 0.0867 | 0.0089 |
| SIDER | 0.1153 | 0.0067 |
| HIV | 0.0370 | 0.0030 |
| MUV | 0.0029 | 0.0000 |
| *PharmaBench* | | |
| CYP3A4 | 0.0102 | 0.0000 |
| CYP2D6 | 0.0106 | 0.0000 |
| CYP2C9 | 0.0057 | 0.0000 |
| LogD | 0.0115 | 0.0001 |
| MLMC | 0.0121 | 0.0006 |
| PPB | 0.0104 | 0.0004 |
| RLMC | 0.0074 | 0.0001 |
| Sol | 0.0645 | 0.0000 |
| HLMC | 0.0145 | 0.0000 |
| *Peptide* | | |
| Long-range Peptide | 0.2697 | 0.0000 |

Table 14: Tokenizer statistics across downstream datasets.

**Notes.** Fallback rate is defined as $\text{fallback\_rate} = \frac{\text{fallback\_tokens}}{\text{final\_tokens}}$, which measures the fraction of tokens produced through recursive fallback decomposition during tokenization. UNK rate denotes the fraction of tokens mapped to the [UNK] symbol.

### A.6 Vocabulary Analysis

To further analyze the properties of the learned fragment vocabulary, we report additional statistics regarding fragment frequency coverage, fragment size distributions, vocabulary stability across vocabulary sizes, chemical diversity analysis, and vocabulary-size sensitivity studies. These analyses provide insight into the robustness of the vocabulary construction procedure, the characteristics of the learned fragment vocabulary, and the effect of vocabulary size on downstream performance.

#### A.6.1 Fragment Frequency Coverage

Figure 6 illustrates the cumulative coverage of fragment occurrences when fragments are sorted by decreasing corpus frequency. The distribution is highly skewed: a small number of fragments accounts for the majority of fragment occurrences across the corpus. In particular, the top-ranked fragments rapidly accumulate coverage, while the remaining fragments contribute only marginally. This behavior indicates that the learned vocabulary captures common structural motifs that frequently appear across molecules. Such a distribution is consistent with patterns observed in subword tokenization methods, where token frequencies typically follow a heavy-tailed distribution. The result suggests that a compact vocabulary is sufficient to represent most molecular structures encountered during training.

#### A.6.2 Fragment Size Distribution

Figure 7 shows the frequency-weighted distribution of fragment sizes in the learned vocabulary, where fragment size is measured by the number of atoms contained in each fragment and frequencies correspond to

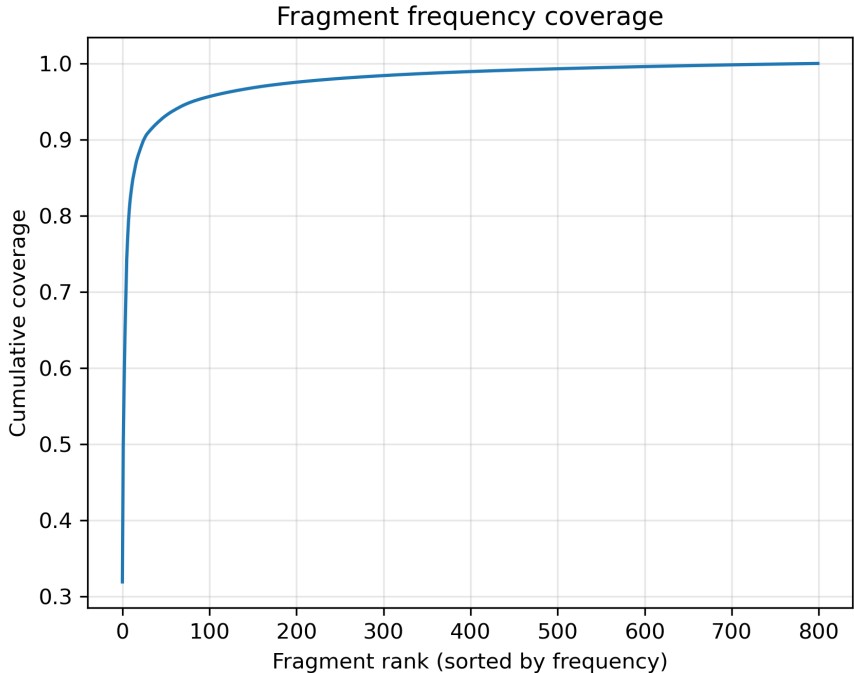

Figure 6: Cumulative coverage of fragment occurrences when fragments are sorted by corpus frequency. A small number of fragments accounts for the majority of occurrences, indicating that the learned vocabulary captures common structural motifs.

the occurrence counts accumulated during vocabulary construction. The distribution is dominated by small fragments, particularly atom-level and short-range motifs, which occur frequently across diverse molecular structures. Larger fragments appear less frequently but capture recurring higher-order chemical patterns such as functional groups and ring systems.

Importantly, this distribution reflects the occurrence statistics of fragments within the learned vocabulary rather than the fragment composition of tokenized molecules. Therefore, the prevalence of small fragments in Figure 7 should not be interpreted as indicating that molecules are represented primarily by atom-level tokens during downstream tokenization. Instead, the figure demonstrates the long-tailed nature of the learned fragment vocabulary.

### A.6.3 Vocabulary Size Analysis

To evaluate the sensitivity of Graph-BPE vocabulary construction to the target vocabulary size, we generated vocabularies containing 400, 800, and 1600 fragments using the same ChEMBL corpus and vocabulary construction procedure described in Section 3.1.

**Chemical Diversity Analysis**  To assess the chemical diversity captured by the learned vocabularies, each fragment was analyzed using a library of 32 SMARTS-based functional-group patterns. The number of unique functional-group categories is defined as the number of distinct SMARTS patterns matched by at least one fragment in the vocabulary. We report both the number of chemically valid fragments after validity filtering and the number of unique functional-group categories represented in each vocabulary.

As shown in Table 15, increasing the vocabulary size substantially increases the number of valid fragments. However, the number of unique functional-group categories grows only modestly from 25 to 28. This suggests that larger vocabularies primarily introduce increasingly rare structural variants rather than substantially expanding the chemical diversity represented by the vocabulary.

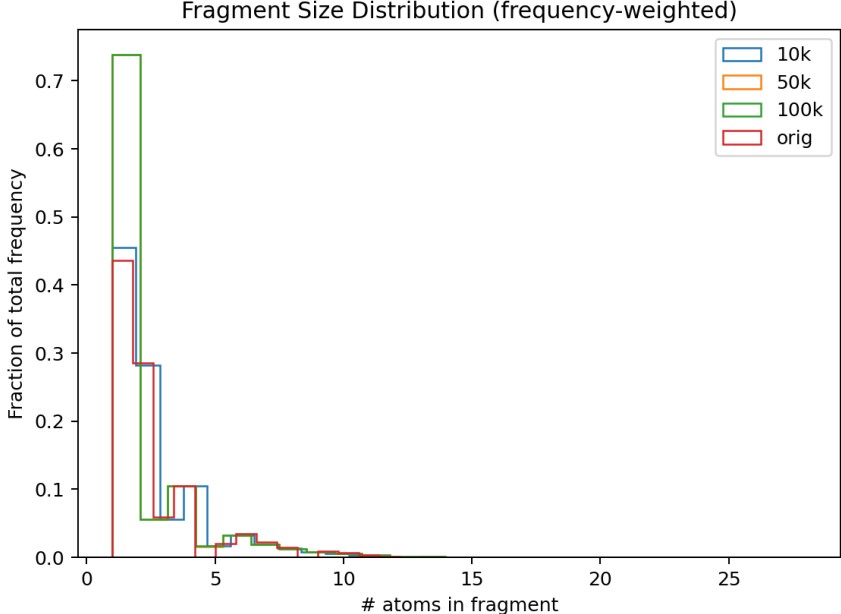

Figure 7: Frequency-weighted distribution of fragment sizes measured by the number of atoms per fragment. Curves correspond to vocabularies learned from different corpus sizes of the same dataset (10k, 50k, 100k molecules), while *orig* denotes the full ChEMBL corpus used for vocabulary construction (430k molecules). Across corpus sizes, most fragments remain small substructures, while larger fragments correspond to recurring chemical motifs captured by the vocabulary.

Table 15: Chemical diversity statistics for different vocabulary sizes.

| Vocabulary Size | Valid Fragments | Functional Group Categories |
|---|---|---|
| 400 | 331 | 25 |
| 800 | 677 | 27 |
| 1600 | 1374 | 28 |

**Vocabulary Size Sensitivity** To evaluate the impact of vocabulary size on downstream performance, we pretrained and fine-tuned BiScale-GTR (Molecule) using vocabularies of size 400, 800, and 1600 while keeping all other training settings unchanged.

Table 16: Downstream performance for different vocabulary sizes.

| Vocabulary Size | Tox21 ROC-AUC | HIV ROC-AUC | Peptides-func AP |
|---|---|---|---|
| 400 | $74.9 \pm 0.3$ | $78.0. \pm 0.8$ | $0.6571 \pm 0.0139$ |
| 800 | $76.1 \pm 0.4$ | $79.2 \pm 0.6$ | $0.6717 \pm 0.0107$ |
| 1600 | $76.3 \pm 0.3$ | $78.9 \pm 0.4$ | $0.6721 \pm 0.0201$ |

Table 16 shows that increasing the vocabulary size from 400 to 800 fragments consistently improves downstream performance across all evaluated datasets. Specifically, the 800-fragment vocabulary improves Tox21 ROC-AUC from 74.9 to 76.1, HIV ROC-AUC from 78.0 to 79.2, and Peptides-func AP from 0.6571 to 0.6717. In contrast, further increasing the vocabulary size to 1600 yields negligible performance changes despite more than doubling the number of valid fragments (677 to 1374) and increasing the number of represented functional-group categories by only one (27 to 28). These results suggest that most chemically

relevant motifs are already captured by the 800-fragment vocabulary, while larger vocabularies primarily introduce additional fragment variants with limited benefit for downstream prediction. Therefore, we adopt the 800-fragment vocabulary in the main experiments as a favorable trade-off between chemical diversity, downstream performance, and vocabulary construction cost.

### A.7 Ablation on Tokenization Refinements.

Table 17 evaluates the effect of the proposed tokenization refinements on the LRGB benchmark. Long-range molecules contain diverse structural motifs, leading to more frequent fallback during tokenization. We compare our full tokenizer with a Graph-BPE baseline that does not include chemical validity filtering and the OOV fallback decomposition mechanism. As shown in Table 17, incorporating these refinements improves performance on both tasks, increasing AP from 0.6321 to 0.6717 on Peptides-func and reducing MAE from 0.2929 to 0.2621 on Peptides-struct. These results highlight the importance of chemical validity filtering and OOV fallback decomposition.

Table 17: Effect of tokenization refinements on long-range molecular datasets. Removing the validity filtering and OOV fallback mechanisms leads to degraded performance.

| Method | Peptides-func (AP) | Peptides-struct (MAE) |
| --- | --- | --- |
| Graph BPE (without validity/OOV handling) | 0.6321 | 0.2929 |
| Graph BPE (with validity/OOV handling) | 0.6717 | 0.2621 |

### A.8 Ablation of Structure-Aware Attention Biases

To evaluate the contribution of the proposed structure-aware attention mechanism, we individually remove the adjacency bias ($B_{adj}$), shortest-path distance bias ($B_{dist}$), and bond structural bias ($B_{bond}$) from the Transformer attention computation. For each ablation variant, the model is pretrained from scratch using the modified attention mechanism and then fine-tuned on the same downstream datasets using the same training protocol as the full model. Table 18 evaluates the contribution of the three structure-aware attention biases.

Table 18: Ablation of structure-aware attention biases.

| Variant | BBBP | Tox21 | MUV | BACE | ToxCast | SIDER | HIV |
| --- | --- | --- | --- | --- | --- | --- | --- |
| Full model | 68.4±0.8 | 76.1±0.4 | 81.6±0.9 | 85.0±1.1 | 62.2±0.9 | 64.2±0.9 | 79.2±0.6 |
| w/o $B_{adj}$ | 67.4±0.8 | 75.2±0.4 | 79.8±0.7 | 83.3±0.9 | 61.5±0.3 | 63.1±0.8 | 78.3±0.4 |
| w/o $B_{dist}$ | 67.0±0.7 | 74.1±0.6 | 79.1±0.7 | 82.1±0.9 | 61.3±0.5 | 62.2±0.7 | 77.1±0.5 |
| w/o $B_{bond}$ | 68.3±0.9 | 75.9±0.5 | 80.9±0.8 | 84.1±1.2 | 62.1±0.8 | 63.4±0.8 | 78.4±0.4 |
| w/o all biases | 65.8±0.8 | 73.8±0.8 | 77.9±0.8 | 80.6±0.8 | 60.5±0.4 | 61.4±0.8 | 77.6±0.6 |

Removing any individual bias consistently reduces performance across all datasets, indicating that each structural signal contributes to the effectiveness of the Transformer encoder. Among the three components, removing the shortest-path distance bias ($B_{dist}$) results in the largest performance degradation on most benchmarks, followed by the adjacency bias ($B_{adj}$). This suggests that explicitly encoding fragment-level topological relationships helps the Transformer better capture molecular structure. In contrast, removing the bond structural bias ($B_{bond}$) leads to smaller but still consistent decreases in performance, indicating that bond-type information provides complementary structural cues. Finally, removing all structural biases produces the largest performance drop across all datasets. These results demonstrate that the three bias terms provide complementary information and that incorporating molecular topology directly into the attention mechanism improves fragment-level representation learning.

Table 19: Effect of GNN depth (ROC-AUC %).

| Variant | BBBP | Tox21 | MUV | BACE | ToxCast | SIDER | HIV |
|---|---|---|---|---|---|---|---|
| 3-layer GNN (Full model) | 68.4±0.8 | 76.1±0.4 | 81.6±0.9 | 85.0±1.1 | 62.2±0.2 | 64.2±0.9 | 79.2±0.6 |
| 6-layer GNN | 62.9±0.9 | 76.2±0.3 | 78.2±0.7 | 81.7±0.8 | 61.3±0.3 | 59.8±0.6 | 79.1±0.4 |
| 8-layer GNN | 60.2±0.7 | 75.3±0.6 | 77.8±0.8 | 79.1±0.9 | 61.1±0.5 | 59.6±0.4 | 78.3±0.7 |

Table 20: Effect of masking ratio on the masked fragment prediction (MFP) pretraining objective.

| Mask Ratio | MFP Accuracy (%) ↑ | MFP Loss ↓ |
|---|---|---|
| 10% | 76.8 | 0.8740 |
| 20% | 70.2 | 1.037 |
| 30% | 70.9 | 0.9991 |

### A.9 Ablation of GNN Depth

Deeper GNNs are known to be prone to overfitting and over-smoothing. Since the GNN in our model is intended to provide a complementary structural prior, we adopt a shallow architecture with 3 layers by default. We further evaluate deeper variants with 6 and 8 layers to examine whether a shallow GNN is sufficient and how GNN depth impacts model performance, as shown in Table 19.

The results show that the 3-layer GNN consistently achieves the best performance across most datasets, while increasing depth leads to performance degradation. This drop is particularly significant on BBBP, suggesting that such datasets require sharper representation boundaries. In this case, deeper GNNs may overfit or over-smooth the representations, thereby blurring decision boundaries and degrading performance. Overall, these findings indicate that a shallow GNN is sufficient for capturing local structural information, while deeper architectures may introduce redundant or noisy representations.

### A.10 Pretraining Objective Analysis

To further analyze the pretraining behavior of BiScale-GTR, we report both the effect of masking ratio and the effect of the proposed frequency-aware masking strategy. Table 20 presents masked fragment prediction (MFP) validation accuracy and loss under different masking ratios. While lower masking ratios yield higher reconstruction accuracy, downstream performance is not strictly correlated with pretraining objective metrics, motivating the use of downstream transfer performance as the primary criterion for selecting the masking ratio.

To evaluate the proposed frequency-aware masking strategy, we additionally compare it against standard uniform masking. As shown in Table 21, frequency-aware masking consistently improves downstream performance across all benchmarks, indicating that prioritizing fragment sampling according to corpus frequency leads to more effective molecular representations.

### A.11 Atom Features

In addition to standard atom attributes, we incorporate a small set of atom-level constraint features derived from the molecular graph to provide additional chemical context for the GNN encoder. For each atom $v_i$, we compute a four-dimensional constraint vector based on its bonding configuration and aromaticity. These features are concatenated with the atom embeddings before message passing. The constraint features are summarized in Table 22.

### A.12 Attention Rollout for Fragment Importance

We estimate fragment-level importance using an attention rollout method that aggregates self-attention weights across Transformer layers. For each layer, multi-head attention weights are first averaged across

Table 21: Effect of masking strategy during pretraining. The full model uses the proposed frequency-aware masking strategy, while Uniform replaces it with standard random masking. Results for the Uniform baseline are averaged over three independent runs.

| Variant | BBBP | Tox21 | MUV | BACE | ToxCast | SIDER | HIV |
|---|---|---|---|---|---|---|---|
| Frequency-aware masking (Full) | 68.4 | 76.1 | 81.6 | 85.0 | 62.2 | 64.2 | 79.2 |
| Uniform masking | 67.2 | 75.3 | 80.1 | 83.1 | 61.9 | 63.7 | 77.4 |

Table 22: Atom-level constraint features used by the GNN encoder.

| Feature | Description |
|---|---|
| Max valence | Maximum valence of the atom as determined by RDKit |
| Bond order sum | Sum of bond orders of all bonds connected to the atom |
| Remaining valence | Difference between maximum valence and bond order sum |
| Aromatic indicator | Binary flag indicating whether the atom is aromatic |

heads. Residual connections are incorporated by adding the identity matrix followed by row-wise normalization. The overall attention propagation is then computed by recursively multiplying the attention matrices across layers:

$$R = \hat{A}^{(L)} \hat{A}^{(L-1)} \cdots \hat{A}^{(1)}, \tag{6}$$

where $\hat{A}^{(l)}$ denotes the normalized attention matrix at layer $l$. We use the [CLS] token as the global representation, and define the importance of each fragment token $i$ as:

$$s_i = R_{\text{CLS},i}, \tag{7}$$

where $s_i$ denotes the importance score of fragment token $i$, defined as its contribution to the [CLS] representation in the attention rollout matrix. Padding tokens are excluded using the key padding mask. The resulting fragment-level importance scores are mapped back to the atoms within each fragment, enabling visualization on molecular structures.

### A.13 Rank-Normalized Fragment Importance for Visualization

For Figure 3, the color scale shows rank-normalized fragment importance rather than raw attention rollout magnitude. For each molecule, we first compute raw fragment attribution scores using the attention rollout procedure described in Appendix A.12. Padding fragments are excluded using the key padding mask. The remaining valid fragment scores are then ranked within the same molecule. Given $n$ valid fragment tokens in a molecule, the rank-normalized importance of fragment $i$ is defined as:

$$\tilde{s}_i = \frac{\text{rank}(s_i)}{n-1}, \tag{8}$$

where $\text{rank}(s_i)$ denotes the zero-based rank of the raw rollout score $s_i$ among valid fragments in that molecule. Thus, the lowest-ranked valid fragment is assigned a value of 0, the highest-ranked valid fragment is assigned a value of 1, and intermediate fragments are evenly spaced between 0 and 1 according to their rank. Therefore, the color scale in Figure 3 reflects relative importance among fragments within the same molecule, not the raw attribution magnitude or a calibrated biological effect size.

### A.14 Normalized Mutual Information (NMI)

To quantify the alignment between learned fragment embeddings and fingerprint-based structural clusters, we use normalized mutual information (NMI). Given two cluster assignments $X$ and $Y$, NMI is defined as:

$$\text{NMI}(X, Y) = \frac{2I(X;Y)}{H(X) + H(Y)}, \tag{9}$$

where $I(X;Y)$ denotes the mutual information between $X$ and $Y$, and $H(\cdot)$ denotes entropy. Higher NMI values indicate stronger agreement between the two clustering.

