# OpenReview forum: "BiScale-GTR: Fragment-Aware graph Transformers for Multi-Scale Molecular Representation Learning"
_TMLR — Under review for TMLR_

### Review · Reviewer_wC7t · 2026-05-21

**Summary Of Contributions:**

The authors investigate better architecture and training of hybrid GNN-Transformer for molecular property prediction. They introduce BiScale-GTR as a unified framework for self-supervised molecular representation learning, combining chemically grounded tokenization with adaptive multiscale reasoning. The proposed tokenization operates on the fragment level and is alleged to be consistent and chemically valid. The multiscale reasoning is enabled by processing information at both the atom and fragment levels. Experiments on 3 benchmarks, namely MoleculeNet, PharmaBench, and Long Range Graph Benchmark show strong empirical performance on classification and regression tasks. The authors also claim that BiScale-GTR, unlike most existing hybrid GNN-Transformer architectures, can learn meaningful representations that are not dominated by GNN modules.


Strengths
1. The bi-scale encoding of molecules by atom-level GNN encoding and fragment-level Transformer encoding makes sense in general. Figure 1 conveys the idea fairly well.
2. The graph BPE tokenization looks reasonable. With that said, I am not an expert on tokenization, so this comment does not warrant correctness.
3. Quantitative results look decent. Compared to existing methods, BiScale-GTR shows competitive performance on MoleculeNet with (in general) fewer pretraining data. It is also on-par or better than existing methods on PharmaBench and two peptide benchmarks.
4. Ablation results are decently comprehensive and straightforward. For example, Table 4 shows that the hybrid GNN-Transformer architecture suffers from nontrivial decrease in performance when either the GNN or the Transformer component is absent.

Weaknesses
1. The authors made this claim in the abstract: “many existing hybrid architectures remain GNN- dominated, causing the resulting representations to remain heavily shaped by local message passing.” I would appreciate if the authors can show some qualitative or quantitative demonstration to back up this claim.
2. The authors emphasized the design of the “structure-aware attention” but I do not seem to find ablation studies or detailed analyses on that.

**Audience:**

Yes

**Audience Explanation:**

The topic of better representations of molecules should be interesting to quite a few people in TMLR's audience.

**Broader Impact Concerns:**

No concern at the moment.

**Claims And Evidence:**

Yes

**Claims Explanation:**

Overall, the claims are supported by the results, with exceptions: see Weakness #1.

**Requested Changes:**

See weaknesses #1 and #2.

---

> ### Author Response · Authors · 2026-06-29
> **Responses to reviewer wC7t**
>
> Comment 1: We thank the reviewer for pointing this out. We agree that the original phrase ``GNN-dominated'' in the abstract was too broad without direct quantitative evidence. We therefore removed this phrasing from the revised abstract and reframed the motivation around context-grounded shared fragment tokens. The revised manuscript no longer claims that prior hybrid architectures are generally GNN-dominated. Instead, Section 2.1 now more carefully discusses sequential, interleaved, and parallel GNN--Transformer designs, and clarifies that BiScale-GTR explicitly separates atom-level contextualization from fragment-level Transformer reasoning.
>
> Comment 2: We thank the reviewer for this suggestion. We added an ablation study for the structure-aware attention module in Appendix A.8 and Table 18. Specifically, we evaluate variants that remove the adjacency bias ($B_{\mathrm{adj}}$), shortest-path distance bias ($B_{\mathrm{dist}}$), bond structural bias ($B_{\mathrm{bond}}$), and all structural biases. The results show that each structural bias contributes to performance, with the largest degradation generally observed when removing the shortest-path distance bias $B_{\mathrm{dist}}$ or all structural biases.

---

> > ### Comment · Reviewer_wC7t · 2026-07-09
> > **Response to author response**
> >
> > I would like to thank the author for the timely response.
> >
> > Meanwhile, I want to point out a few cosmetic issues:
> >
> > 1. Typo in Table 18: "77.6.6±0.6" is likely "77.6±0.6"
> > 2. Table 1 first column width should be increased to make sure "GraphGPS+LAC (Yang et al., 2025)" stays in the same line without line break.
> >
> > Other than that, my concerns are addressed.

---

> > > ### Author Response · Authors · 2026-07-13
> > >
> > > We thank the reviewer for the prompt comments and careful reading. We have corrected the numerical typo in Table 18 and fixed the formatting issue in Table 1 by increasing the width of the first column. We greatly appreciate the reviewer pointing out these issues.

---

### Review · Reviewer_U7CD · 2026-05-22

**Summary Of Contributions:**

The paper addresses multi-scale molecular representation learning through a hybrid architecture combining Graph Neural Networks (GNNs) and Transformers. The proposed method constructs a fragment vocabulary via a graph-based Byte Pair Encoding (BPE) procedure guided by Weisfeiler–Lehman (WL) hashing. Atom-level node embeddings are learned through the GIN and subsequently aggregated via attention pooling to derive fragment-level representations. In parallel, fragment token embeddings are obtained directly from a learned embedding table. These two embeddings are integrated through a gated fusion mechanism to form the final fragment representation that serves as input to the Transformer. The Transformer employs structure-aware attention to capture molecular topology at the fragment level. The resulting model, BiScale-GTR, is pretrained on ChEMBL via a masked fragment prediction objective and fine-tuned on downstream tasks spanning MoleculeNet, PharmaBench ADMET, and the LRGB. The paper further includes ablation studies and component-level analyses.


**Strengths**

1. The proposed method is evaluated across a diverse set of tasks using a shared architecture, including classification tasks on MoleculeNet, regression tasks on PharmaBench ADMET, and multi-label classification and regression tasks on the Peptides-func and Peptides-struct datasets from the LRGB benchmark.
2. The paper proposes a gated fusion mechanism to integrate fragment-level token representations with pooled atom-level GNN representations, enabling the model to jointly capture structural information at both the atom level and the fragment level within a unified framework.
3. The model achieves competitive performance compared to baselines pretrained on substantially larger corpora of up to 2M molecules, while using only 430K pretraining molecules, demonstrating notable data efficiency.


**Weaknesses**

1. The motivation for combining GNN-based local encoding with Transformer-based global encoding requires clearer justification. Transformer architectures equipped with appropriate masking strategies are capable of capturing both local and global dependencies, and stacking multiple GNN layers can model long-range interactions. The paper should more clearly state why a hybrid architecture is necessary, and provide empirical or theoretical evidence that neither component alone is sufficient. While the ablation in Table 4 demonstrates that the combination outperforms each component in isolation, it does not explain why this particular fusion design is preferable over alternative hybrid strategies.
2. The authors state that "…many molecular properties are governed by interactions between higher-level structural motifs such as functional groups and substructures (Schaeffer, 2008)." However, the cited reference concerns drug–receptor interactions, and its applicability to the molecular property prediction tasks evaluated is not clear. A similar concern applies to the following citation (Jinsong et al., 2024). The authors should provide references that more directly support these claims.
3. In Figure 3, the maximum importance values are approximately 0.02 and 0.05, respectively. It is unclear whether attribution scores of such small magnitude provide sufficient evidence that the highlighted substructures are genuinely relevant to biological activity. The authors should clarify how the annotation text in the figure was derived, verify that the cited references directly support the specific biological claims.
4. The computational complexity of Algorithm 1 is not discussed. Given that the proposed pipeline introduces multiple components, the authors should report the time and memory complexity of the tokenization procedure, as well as end-to-end training and inference costs relative to baseline methods.
5. The effect of vocabulary size on downstream performance is not ablated. The paper adopts a fixed vocabulary size of 677 valid fragments without investigating the sensitivity of the model to this choice. Furthermore, the limitation arising from constructing the vocabulary from the ChEMBL dataset should be discussed, particularly given the observed fallback rate of approximately 26% on peptides.
6. In Section 3.2.1, the paper states that edge representations are injected into the GIN message-passing layers. However, these edge representations do not appear in Equation (2).
7. The paper introduces three structural biases into the Transformer attention mechanism but does not provide ablation results isolating the contribution of each component. Such ablations are necessary to support the claim that structure-aware attention effectively captures long-range dependencies between fragments. Additionally, it is unclear how bond direction is encoded in the bond-type structural bias.
8. The pretraining analysis in Table 5 reports only downstream task performance under different masking ratios. The authors do not report evaluation metrics on the pretraining objective itself, nor do they provide a direct comparison between the proposed frequency-aware masking strategy and uniform masking. Such analysis is necessary to support the claims made regarding the proposed sampling strategy.
9. In Section 5.1, the authors state that the two model variants are introduced "to study the effect of oversmoothing in graph neural networks." However, the subsequent analysis is framed in terms of local structural reasoning versus long-range interactions, which is a conceptually distinct phenomenon from oversmoothing. The relationship between these concepts is not clearly established, and it is not clear how differences in ROC-AUC between the two variants can be specifically attributed to oversmoothing rather than other architectural differences.
10. In Section 5.3, the authors suggest that the performance gap on Peptides-struct is partially explained by the absence of 3D geometric information in the proposed model. However, it is not clearly stated which baseline methods in the comparison incorporate 3D geometric information.
11. The faithfulness evaluation and attention attribution analysis are conducted exclusively on classification tasks. Extending this analysis to regression tasks would provide a more complete assessment of the model's interpretability across different task types.
12. The t-SNE visualizations in Figure 4 are difficult to interpret. The embedding distribution in Figure 4(b) appears visually more dispersed than Figure 4(a), which is inconsistent with the authors' claim of tighter and more separable clusters.
13. In the conclusion, the authors state that "leveraging 3D information introduces practical challenges, including the limited availability of high-quality structures and the need for reliable conformation generation." This statement is inaccurate, given the existence of several well-established and widely used 3D molecular datasets such as QM9 and GEOM-Drug.

**Audience:**

Yes

**Audience Explanation:**

Understanding how atomic-level structures and functional group-level fragments jointly determine molecular properties is a good question in molecular science. Data-driven approaches that provide interpretable multi-scale representations offer valuable insights into structure–activity relationships, and findings in this direction are of broad interest to researchers in both machine learning and computational chemistry communities.

**Broader Impact Concerns:**

I have no concerns regarding the ethical implications of this work.

**Claims And Evidence:**

No

**Claims Explanation:**

1. The paper states that “…, where accurate property prediction can depend on interactions between distant functional groups (Yang et al., 2019).” However, the cited reference does not appear to provide direct evidence for this claim in the context of the tasks evaluated. More importantly, the experimental results present a challenge to this claim: BiScale-GTR (Molecule) outperforms BiScale-GTR (Fragment) on the majority of MoleculeNet datasets (Table 1), and Tables 2 and 3 report results for BiScale-GTR (Molecule) only, without providing the performance of BiScale-GTR (Fragment) on PharmaBench and LRGB benchmarks. This appears to contradict the statement that fragment-level local motif reasoning is a primary driver of performance. If interactions between functional groups are the dominant factor, the fragment-centric variant, which preserves sharper local token distinctions as shown in Table 8, would be expected to perform more consistently better.
2. The advantages of the proposed graph-based BPE tokenization with WL hashing over simpler alternatives are not clearly demonstrated. As shown in Figure 7, the majority of learned fragments contain only 2–3 atoms, which is close to atom-level granularity and raises questions about whether fragment-level reasoning provides meaningful additional expressiveness. Furthermore, Figure 6 shows that a small number of fragments accounts for the majority of occurrences, which calls into question whether the full complexity of graph BPE with WL hashing is necessary compared to simpler frequent substructure mining approaches. The paper does not include direct comparisons against alternative tokenization strategies to justify this design choice. Additionally, given the potential overlap between ChEMBL and the MoleculeNet and PharmaBench benchmarks, the generalizability of the learned vocabulary to truly out-of-distribution (OOD) molecules is not clearly established. The motivation for requiring a shared vocabulary across molecules, as opposed to molecule-specific fragmentation strategies, is also not explicitly justified.

**Requested Changes:**

- Must have:
W1, W2, W3, W5, W6, W7, W8, W9, W10, and the two unsupported claims

- More suggestions or questions:
W4, W11, W12, W13

---

> ### Author Response · Authors · 2026-06-29
> **Response to Reviewer U7CD: Points 1–5**
>
> 1. We thank the reviewer for this insightful comment. BiScale-GTR is designed around the principle of context-grounded shared fragment tokens: reusable fragment identities should be shared across molecules, while each occurrence should still be adapted to its local atomic environment. In this setting, the GNN and Transformer play distinct roles. Fragment tokenization provides higher-level reusable molecular units, but replacing an atom-level subgraph with a fragment token can abstract away fine-grained atom and bond information. The shallow GNN therefore serves as an atom-level grounding module, injecting bond-constrained local chemical context into each fragment occurrence through atom-to-fragment pooling. The fragment Transformer then performs global reasoning over these atom-grounded fragment tokens. Thus, the architecture is not simply a generic combination of local GNN and global Transformer modules; it is designed to preserve transferable fragment identity while allowing each fragment occurrence to acquire molecule-specific atomic context. We strengthened this justification in the revised manuscript, including the introduction, Section 3.2, and the ablation discussion in Section 5.4. Empirically, Table 4 shows that removing the GNN degrades performance, indicating that reusable fragment tokens and fragment-level self-attention alone do not fully capture occurrence-specific local chemical context. Conversely, the GNN-only variant performs substantially worse, showing that local message passing alone is insufficient for global fragment-level reasoning. The fixed-sum fusion variant also underperforms adaptive fusion, supporting the role of the gate in grounding reusable fragment tokens with atom-derived context. We also revised the ablation discussion to clarify its scope. Rather than claiming superiority over all possible GNN--Transformer hybrids, Table~4 evaluates the role of adaptive atom--fragment fusion within the proposed framework.
> 2. We thank the reviewer for this observation. In the revised manuscript, we removed the overstated statements. Instead, we now frame the motivation around combining reusable fragment-level abstraction with atom-level contextual grounding. We cite Gasteiger et al. (2020) and Stärk et al. (2022) to support the importance of local atomic and electronic context in molecular representation learning.
> 3. We thank the reviewer for this helpful comment. The original Figure 3 displayed raw CLS attention rollout scores, which are naturally small because attention is normalized across fragments and compounded across layers. We revised Figure 3 to use rank-normalized fragment importance within each molecule, so the visualization reflects relative importance rather than small raw attention-rollout values. We also clarified how the annotations were derived: the top-ranked fragments were extracted, matched to functional-group patterns using RDKit, and verified against supporting literature. The revised manuscript retains only citations that directly support the annotated motifs.
> 4. We thank the reviewer for this helpful suggestion. We added tokenization complexity analysis in Appendix A.4 and empirical computational costs in Appendix A.3. The revised manuscript clarifies that vocabulary construction is a one-time offline preprocessing step, while inference-time tokenization is lightweight in practice. We also report absolute runtime and GPU-hour costs, including vocabulary construction, pretraining, fine-tuning, and inference, instead of making uncontrolled wall-clock comparisons against baselines trained under different hardware and settings.
> 5. We thank the reviewer for this helpful suggestion. We have added a vocabulary-size sensitivity study in Appendix A.6, comparing vocabularies constructed with target sizes of 400, 800, and 1600 fragments. The results show that increasing the vocabulary size from 400 to 800 improves downstream performance, while further increasing it to 1600 provides only marginal gains. We also observe that chemical diversity saturates quickly across these vocabulary sizes. Based on the trade-off among downstream performance, chemical coverage, vocabulary complexity, and construction cost, we retain the 800-fragment construction setting. We also expanded the discussion of the ChEMBL-derived vocabulary in Section 3.1.3. Because the vocabulary is learned from ChEMBL small molecules, it is biased toward drug-like substructures and is less likely to contain peptide-specific higher-order motifs such as repeated backbone and residue-level patterns. This domain mismatch explains the higher fallback rate of approximately 26% on the LRGB peptide datasets. We now explicitly discuss this as a limitation of using a fixed small-molecule-derived vocabulary. At the same time, the strong performance on Peptides-func and competitive performance on Peptides-struct suggest that recursive decomposition still provides useful coverage under this structural shift.

---

> ### Author Response · Authors · 2026-06-29
> **Response to Reviewer U7CD: Points 6–12**
>
> 6. We thank the reviewer for pointing this out. The original Equation (2) showed the standard GIN update and omitted the edge embeddings used in our implementation. We revised Section 3.2.1 to describe the encoder as an edge-aware GIN (GINE) and updated Equation (2) to include the edge embedding term $e_{ij}$, which encodes bond type and bond direction information. This revision ensures consistency between the mathematical formulation and the implementation.
> 7. We thank the reviewer for this suggestion. We have added ablation results for the three structure-aware attention biases in Appendix A.8 and Table 18 by removing $B_{\mathrm{adj}}$, $B_{\mathrm{dist}}$, $B_{\mathrm{bond}}$, and all structural biases. The results show that each component contributes to performance, with the largest drop generally observed when removing $B_{\mathrm{dist}}$ or all structural biases, supporting the importance of shortest-path information for modeling fragment-level relationships. Second, we clarified the bond-direction encoding in Section 3.2.4: for adjacent fragment pairs, bond type and RDKit-derived bond direction are encoded by separate learnable embeddings and added as the bond-specific attention bias.
> 8. We thank the reviewer for this helpful suggestion. We have added additional pretraining analysis in Appendix A.10. Specifically, we now report masked fragment prediction accuracy and validation loss under different masking ratios in Table 20, directly evaluating the pretraining objective itself. The results show that 10% masking gives the easiest pretraining objective, with the highest MFP accuracy and lowest validation loss, but does not yield the best downstream transfer. In contrast, 20% masking achieves the strongest downstream performance across most benchmarks, suggesting that objective accuracy alone is not sufficient for selecting the masking ratio. We also added a direct comparison between the proposed frequency-aware masking strategy and standard uniform masking in Table 21. Under the same model and training settings, frequency-aware masking consistently improves downstream performance across the evaluated MoleculeNet benchmarks, including clear gains on MUV, BACE, and HIV. These results support the use of a moderate masking ratio and the proposed frequency-aware masking strategy.
> 9. We thank the reviewer for this observation. We agree that the original wording was imprecise. Our intention was not to directly study oversmoothing, but rather to examine the impact of GNN message-passing scope by comparing fragment-level and full-molecule GNN encoding. We have revised Section 5.1 accordingly, replacing the oversmoothing-related statement and clarifying that the observed performance differences are discussed in terms of the amount of structural context available to each variant rather than as direct evidence of oversmoothing.
> 10. We thank the reviewer for this observation. We agree that the original discussion could be misinterpreted because the compared baselines also do not incorporate explicit 3D geometric information. Since the statement was not essential to our analysis, we have removed it from the revised manuscript and focused the discussion on the empirical comparison between methods.
> 11. We thank the reviewer for this helpful suggestion. We agree that extending interpretability analysis to regression tasks would provide a more complete assessment across task types. In this work, we focus the faithfulness evaluation on classification benchmarks because our perturbation protocol measures ROC-AUC degradation after removing highly attributed fragments, which provides a direct and comparable metric across classification datasets. For regression tasks, an analogous protocol would require a different faithfulness metric, such as prediction shift or MAE/RMSE change after fragment removal. However, these quantities are endpoint-dependent and not directly comparable across regression tasks with different units, value ranges, and biological meanings. We leave a systematic regression-task interpretability study with regression-specific perturbation metrics for future work.
> 12. We thank the reviewer for pointing this out. Figure 4 already shows stronger same-color grouping in the Transformer+GNN setting than in the Transformer-only setting. To avoid relying only on visual inspection, we further report NMI between learned fragment embeddings and ECFP-derived structural clusters. The Transformer+GNN model achieves a higher NMI than the Transformer-only model (0.7678 vs. 0.7009), confirming stronger structural alignment. We revised the text to clarify that the claim refers to improved within-cluster grouping and structural alignment, rather than global compactness of the entire t-SNE projection.

---

> ### Author Response · Authors · 2026-06-29
> **Response to Reviewer U7CD: Points 13-Claim 1**
>
> 13. We thank the reviewer for this clarification. We agree that the original wording was too broad. Our intended point was not that 3D molecular data are generally unavailable, but that explicit 3D geometry is not consistently provided as standard input across the diverse benchmarks considered in this work. Incorporating 3D information would therefore require additional choices. We have revised the conclusion to explicitly acknowledge available large-scale 3D resources and to clarify that this work focuses on 2D molecular graphs in order to isolate the effect of fragment-aware multi-scale representation learning.
>
> Claim1:  We thank the reviewer for this comment. We agree that the original statement about interactions between distant functional groups was too strong and not directly supported by the cited reference. We revised the wording to: ``This limitation is particularly relevant for molecular graphs, where molecular property prediction can require global molecular information beyond local message passing.'' This revised statement better reflects the evidence provided by Yang et al. and avoids overclaiming that distant functional-group interactions are the dominant factor. We would like to clarify the role of the two BiScale-GTR variants in Section 5.1. BiScale-GTR (Fragment) and BiScale-GTR (Molecule) are not comparisons between fragment-level and non-fragment-level reasoning; both variants operate on fragment tokens and perform Transformer reasoning at the fragment level. The difference lies in the scope of atom-level GNN encoding before atom-to-fragment pooling. The Fragment variant preserves isolated local fragment semantics, while the Molecule variant provides context-aware fragment representations by encoding the full molecular graph. Thus, the stronger performance of BiScale-GTR (Molecule) on most datasets does not contradict the importance of fragment-level reasoning; rather, it suggests that many tasks benefit from atom-grounded fragment tokens that incorporate broader molecular context. We therefore use BiScale-GTR (Molecule) as the default full model for PharmaBench and LRGB, while BiScale-GTR (Fragment) serves as a diagnostic variant for analyzing the local-versus-contextual representation trade-off.

---

> ### Author Response · Authors · 2026-06-29
> **Response to Reviewer U7CD: Claim 2**
>
> We thank the reviewer for raising these concerns. We agree that the original manuscript did not sufficiently justify the proposed tokenizer against simpler alternatives, and we have added a controlled tokenizer ablation study in Section 5.4 and Table 5 to address this directly. The ablation compares atom-level tokenization, BRICS tokenization, Graph-BPE with SMILES-based identity, and the proposed Graph-BPE + WL tokenizer under the same BiScale-GTR architecture, pretraining corpus, optimization settings, and fine-tuning protocol. Graph-BPE + WL achieves the strongest overall performance and clearly outperforms atom-level tokenization on several benchmarks, including BBBP, BACE, SIDER, and Peptides-func. This provides direct empirical evidence that the proposed graph-aware fragment tokenizer provides useful representational structure beyond atom-level inputs.
>
> Regarding fragment size, we agree that many learned fragments are compact substructures. However, compact fragments are not equivalent to atom-level tokenization. A 2--3 atom fragment can encode local bond patterns and recurring chemical environments that are not represented when each atom is treated as an independent token. Moreover, Figure 7 reports the frequency-weighted distribution of learned vocabulary entries; because small chemical primitives are reused across many molecules, they naturally dominate the frequency distribution. We therefore rely on controlled downstream ablations, rather than fragment-size distribution alone, to justify the tokenizer.
>
> Regarding Figure 6, the long-tailed occurrence distribution is expected for BPE-style vocabularies: a small number of common fragments provides broad coverage, while less frequent fragments provide additional structural specificity. Frequency coverage alone is not the only purpose of the tokenizer. Compared with generic frequent substructure mining, Graph-BPE produces a deterministic, non-overlapping tokenization of each molecule, records a merge history for recursive decomposition, and uses WL hashing to assign permutation-invariant identities to isomorphic fragments. The comparison with BRICS and Graph-BPE(SMILES) in Table\~5 further supports this design choice: improvements over BRICS suggest that learned graph-based fragments capture useful recurring motifs beyond fixed rule-based fragmentation, while improvements over Graph-BPE(SMILES) show the benefit of graph-invariant WL identity, chemical validity filtering, and recursive OOV handling.
>
> Regarding generalization beyond ChEMBL-like small molecules, we agree that MoleculeNet and PharmaBench are largely drug-like and may be structurally closer to ChEMBL. We therefore expanded the discussion in Section 3.1.3 using the LRGB peptide datasets as a structurally distinct test case. The higher fallback rate on peptides, approximately 26 %, indicates that many peptide structures are outside the ChEMBL-derived vocabulary, and we explicitly ackowledge this as a limitation of using a fixed small-molecule-derived vocabulary. Nevertheless, BiScale-GTR achieves the best reported performance on Peptides-func and competitive performance on Peptides-struct, suggesting that the learned vocabulary and recursive decomposition remain useful under this structural shift.
>
> Finally, we revised the manuscript to make the motivation for a shared vocabulary more explicit in Sections 2.2, 2.3, and 5.5. Molecule-specific fragmentation can produce valid local subgraphs, but the resulting fragment nodes are not necessarily aligned across molecules. A shared vocabulary assigns consistent token identities to isomorphic fragments across molecules, enabling parameter sharing across recurring motifs, masked fragment pretraining over a consistent token space, and fragment-level attribution over recurring chemical substructures rather than molecule-specific artifacts.

---

> > ### Comment · Reviewer_U7CD · 2026-07-11
> >
> > I thank the authors for their detailed responses and revisions. However, several concerns remain insufficiently addressed.
> >
> > 1. While the revised manuscript provides a clearer description of the roles of the GNN and Transformer components, the motivation remains inadequately supported. The newly cited references (Gasteiger et al., 2020; Stärk et al., 2022) focus on 3D molecular graphs and do not directly support the claims in this work. Furthermore, the boundaries between the three lines of related work discussed — graph Transformers, fragment-based methods, and atom-fragment hybrid approaches — remain insufficiently distinguished, making it difficult to clearly identify the gap that BiScale-GTR addresses.
> >
> > 2. The revision does not clarify how the annotations in Figure 3 were derived, and the cited references do not appear to directly support the specific biological claims made. Furthermore, the highlighted fragments are large in size and likely correspond to low-frequency entries in the learned vocabulary. It remains unclear whether these fragments are consistent with the proposed fragment mining method, and a quantitative analysis would be more convincing than qualitative visualization alone.
> >
> > 3. Figure 4 remains difficult to interpret visually. Regarding the NMI score, using the fingerprint-based clusters as the reference for evaluating the alignment of learned fragment embeddings introduces a potential circularity, given that the proposed fragment mining method is claimed to outperform alternative fragmentation strategies.
> >
> > 4. The discussion of the 3D limitation remains vague. While the authors attribute the lack of 3D evaluation to inconsistent annotations across datasets, it is unclear what specific challenge this poses for the proposed method, especially given that benchmarks such as MoleculeNet have been widely used for 3D models.
> >
> > 5. The justification for the two variants remains unclear. In BiScale-GTR (Molecule), the GNN processes the entire molecular graph, allowing information to propagate across fragment boundaries prior to Transformer reasoning. It is not clear how this design is consistent with the stated motivation of the proposed architecture.

---

> ### Author Response · Authors · 2026-07-13
> **Response to Reviewer U7CD**
>
> We thank the reviewer for their meticulous review and helpful comments. Our responses regarding each comment are made as follows:
> 1. Thank you for commenting on these two references (Gasteiger et al., 2020; Stärk et al., 2022). We have removed this claim and no longer use these references to motivate context grounding. In the revised manuscript, we substantially rewrote the Introduction and Related Work to better support the motivation and distinguish the three relevant lines of work.
> 2. If by "annotations" you refer to Fig. 3, there are three different components. (1) The node colors indicate atoms belonging to the same learned Graph-BPE fragment. (2) The circled fragment is automatically selected as the highest-attribution learned fragment according to the CLS attention rollout procedure described in Appendix A.12. (3) The text labels are manual chemical annotations describing the functional group present in the selected fragment; these were assigned by inspecting the fragment structure and verified against the cited literature. The circled fragments in Fig. 3 contain only 2–6 atoms (including hydrogens), placing them on the smaller side of our learned fragment-size distribution rather than representing unusually large or rare fragments. More generally, our graph-BPE tokenizer learns reusable vocabulary tokens through a frequency-based mining procedure, but the attribution score during inference is independent of fragment frequency. Therefore, even if a highly attributed fragment were larger, this would remain fully consistent with our fragment mining method as long as it is a learned vocabulary token. Finally, we have revised the biological discussion to make it more conservative and avoid implying causal or mechanistic conclusions beyond the supporting literature.
>
> 3. We thank the reviewer for the comment. Fig.4(b) shows stronger within-cluster grouping of fragments than Fig.4(a). For example, the green and red clusters are more dispersed in Fig.4(a), whereas they form more coherent groups when atom-level GNN grounding is incorporated in Fig.4(b). We further improved the caption to make it easy to understand the visualization. The NMI analysis is not circular in the training or evaluation sense. ECFP-based clusters are not used in graph-BPE vocabulary construction, pretraining, fine-tuning, or embedding learning. They are used only post hoc as an external structural reference for assessing whether learned fragment embeddings are organized consistently with conventional chemical similarity. We also clarify that the NMI score is not used to claim that our fragment mining method outperforms alternative tokenizers. That claim is supported separately by the controlled tokenizer ablation studies.
> 4. We clarify that incorporating 3D information is outside the scope of the present work. BiScale-GTR is designed to study context-grounded reusable fragment tokens in a 2D molecular graph setting, where fragment identities are defined by graph-BPE and WL-based graph identity. Incorporating 3D geometry would require additional design choices, including conformer generation/selection and geometry-aware atom-to-fragment grounding. We have revised the limitation discussion to make this scope clear and leave 3D extensions to future work.
> 5. We clarify that the two variants differ in the scope of atom-level grounding, not in the underlying design principle. The motivation of BiScale-GTR is not to enforce strict isolation of fragments before Transformer reasoning; rather, it is to preserve reusable fragment-token identities while grounding each token occurrence with atom-level chemical context. In BiScale-GTR (Fragment), the GNN operates on isolated tokenizer-defined fragment subgraphs, so each fragment token is grounded only by its internal atoms. In BiScale-GTR (Molecule), the GNN operates on the full molecular graph before atom-to-fragment pooling, so each fragment token can additionally incorporate its local attachment and neighboring bond environment. This is consistent with our motivation because the shared fragment identity is still defined by the learned graph-BPE vocabulary, while the pooled atom representation provides occurrence-specific context for that token.
> The fragment Transformer then serves a distinct role: it performs global reasoning over the resulting atom-grounded fragment tokens. Thus, full-molecule GNN grounding does not replace fragment-level Transformer reasoning; it enriches the local context used to ground each reusable token before long-range fragment-level interactions are modeled. We also revised the method description to clarify that BiScale-GTR (Fragment) and BiScale-GTR (Molecule) are diagnostic variants for studying the trade-off between strictly fragment-local grounding and broader molecule-context grounding.

---

### Review · Reviewer_tErT · 2026-06-24

**Summary Of Contributions:**

This paper proposes BiScale-GTR, a molecular property prediction framework that combines atom-level GNN representations with fragment-level Transformer reasoning. The method first constructs a fragment vocabulary using a graph-BPE tokenizer enhanced with WL-hash canonicalization, chemical validity filtering, and an OOV decomposition mechanism, then pools atom representations into fragment embeddings and fuses them with learned fragment tokens before Transformer processing. Experiments on MoleculeNet, PharmaBench, and LRGB benchmarks show competitive performance against recent GNN-, Transformer-, and fragment-based molecular learning methods.

**Additional Comments:**

While the empirical results are generally solid, I find the technical novelty limited. The proposed framework largely combines existing ideas from fragment-based molecular learning, hierarchical graph representation learning, graph tokenization, and GNN–Transformer hybrids. Not only each part is proposed, but also some papers have tried to combine them. The paper demonstrates that these components can work well together, but it does not clearly establish a new methodological insight or principle that substantially advances the state of the art. Consequently, I am not convinced that the level of novelty in the current manuscript is enough for publication.

**Audience:**

Yes

**Audience Explanation:**

The proposed WL-hash canonicalization, chemical validity filtering, and OOV fallback mechanism improve the robustness of graph-BPE tokenization and may be useful engineering improvements for fragment-based molecular modeling.

**Claims And Evidence:**

Yes

**Claims Explanation:**

The paper evaluates the proposed method across multiple benchmark suites, including MoleculeNet, PharmaBench, and LRGB, providing evidence that the approach is generally competitive across classification and regression tasks.

**Requested Changes:**

## 1. Limited technical novelty; the method appears largely as a combination of existing components

The overall framework primarily combines several well-established ideas:

Fragment-based molecular representations have been extensively studied in methods such as GraphFP [1].

Hybrid GNN + Transformer architectures are already common in graph learning, e.g., GraphGPS [2], GraphTrans [3], etc.

Multi-scale molecular representations that combine atom-level and higher-level structural representations have also been widely explored in molecular learning.

Conceptually, the proposed pipeline follows a familiar recipe: fragment decomposition → atom-level encoding → fragment-level aggregation → Transformer reasoning → fusion for prediction. The paper does not clearly identify a fundamentally new modeling principle beyond integrating these existing techniques into a unified architecture. In particular, the core claim of "multi-scale reasoning" appears incremental given the extensive prior literature on hierarchical graph representations and fragment-aware molecular learning.

## 2. Insufficient evidence that the proposed multi-scale design is necessary

A central motivation is that molecular properties depend on both local atom-level information and higher-level fragment interactions. However, the paper does not convincingly demonstrate that explicit fragment-level reasoning provides benefits beyond existing graph-transformer architectures. The ablations mainly remove entire modules (GNN or Transformer) but do not isolate the contribution of the proposed multi-scale hierarchy itself.

## 3. The motivation regarding "single-granularity" molecular representation learning and "GNN-dominated"  appear somewhat overstated.

The manuscript argues that most existing methods operate only at the atom level and therefore fail to capture patterns across multiple molecular scales. However, a substantial body of prior work has already explored multi-scale molecular representations through fragment-based modeling, hierarchical graph learning, graph coarsening, and hybrid GNN-Transformer architectures. In fact, several such methods are already discussed or used as baselines in the paper. Consequently, it is unclear whether the identified limitation truly reflects a gap in the literature, or whether the proposed method primarily represents another instantiation of an already well-established multi-scale representation paradigm. The paper would benefit from a more precise positioning of its novelty relative to these existing approaches.

---

> ### Author Response · Authors · 2026-06-29
> **Response to Reviewer tErT on novelty and positioning**
>
> Comment 1: We thank the reviewer for raising this important point. We agree that fragment-based molecular representations, GNN--Transformer hybrids, and multi-scale molecular modeling have each been studied in prior work. However, BiScale-GTR is not intended as a generic combination of these components. Its contribution is a specific representation principle: context-grounded shared fragment tokens. That is, recurring fragments should share reusable token identities across molecules, while each token occurrence should be grounded with molecule-specific atom-level context before fragment-level global reasoning.
>
> This principle addresses a gap between two related but distinct lines of work. Fragment-vocabulary methods learn reusable substructure identities, but shared token identity alone does not specify how a fragment is instantiated in a particular atomic environment. Atom--fragment hierarchical methods integrate atom- and fragment-level information, but typically use fixed or molecule-specific fragment nodes rather than reusable corpus-level tokens. BiScale-GTR connects these directions by constructing reusable graph-BPE fragment tokens with WL-based matching, chemical validity filtering, and recursive OOV decomposition, and then grounding each token occurrence with atom-level GNN representations through atom-to-fragment pooling and adaptive fusion.
>
> The revised manuscript makes this distinction explicit in the abstract, introduction, and related work. We no longer frame the contribution as simply introducing another fragment model or another GNN--Transformer hybrid. Instead, we position BiScale-GTR as a framework for robust shared fragment tokenization with occurrence-specific atomic grounding before structure-aware fragment-level Transformer reasoning.
>
> Comment 2: We agree that a simple module-removal study alone would not fully establish the necessity of the proposed multi-scale design. We therefore strengthened the revised manuscript with controlled ablations that isolate the roles of representation granularity, atom-level grounding, and fragment-level global reasoning.
>
> The tokenizer ablation evaluates the representation-unit choice under the same architecture and training protocol, comparing atom-level tokenization, BRICS, Graph-BPE with SMILES-based identity, and Graph-BPE with WL-based matching. The results show that explicit fragment-level tokenization improves over atom-level inputs on several benchmarks and that the proposed graph-aware tokenizer achieves the strongest overall performance. This supports the use of reusable fragment units rather than treating molecules only as atom-level graphs.
>
> The architecture ablation further separates the roles of the two scales. The Transformer-only variant retains reusable fragment tokens and fragment-level self-attention but removes atom-level grounding, and its degradation shows that shared fragment identity alone is insufficient. The GNN-only variant removes fragment-level global reasoning and performs substantially worse, showing that local message passing alone is not enough. The fixed-sum fusion variant keeps both sources of information but removes adaptive balancing, and its weaker performance supports the need to adaptively combine shared token identity with atom-derived context.
>
> Together, these results support the proposed design: reusable fragment tokens provide transferable higher-level units, atom-level grounding restores local chemical context for each occurrence, and the fragment Transformer models global interactions among atom-grounded tokens.
>
> Comment 3: We thank the reviewer for this comment. We agree that the original wording could have been read as suggesting that prior work lacks multi-scale or hybrid molecular modeling in general. This was not our intended claim. Prior work has indeed explored fragment-aware models, hierarchical molecular representations, graph coarsening, and GNN--Transformer hybrids. We have revised the manuscript to make the contribution more precise.
>
> Specifically, we revised the Abstract and Section 1 to no longer frame BiScale-GTR as simply addressing the absence of multi-scale molecular modeling. Instead, the revised motivation centers on context-grounded shared fragment tokens: reusable fragment identities that are shared across molecules but grounded with atom-level context for each occurrence. We also revised Section 2.1 to clarify the relationship to existing GNN--Transformer hybrids, Section 2.2 to better position our robust reusable fragment tokenization relative to prior fragment-based methods, and added a dedicated Section 2.3 on atom--fragment molecular representation to discuss HiMol, HimGNN, and FragNet. Thus, the revised manuscript no longer positions BiScale-GTR as the first multi-scale molecular representation model, but as a specific framework for context-grounded shared fragment tokens.

---

> ### Author Response · Authors · 2026-07-16
> **General Update**
>
> As we are approaching the end of the rebuttal period, we would like to note that we have submitted a substantially revised version of the manuscript. In particular, we extensively rewrote the Introduction and Related Work sections to provide a clearer research motivation and more carefully position our work with respect to existing fragment-based, hierarchical, and hybrid GNN–Transformer approaches. As many of these revisions were made in direct response to your comments, we would greatly appreciate any additional feedback you may have after reviewing the updated manuscript.